

# Soil smoldering in temperate forests : A neglected contributor to fire carbon emissions revealed by atmospheric mixing ratios

Lilian Vallet[1], Charbel Abdallah[2], Thomas Lauvaux[2], Lilian Joly[2], Michel Ramonet [3], Philippe Ciais[3], Lopez Morgan[3], Irène Xueref-Remy[4], Mouillot Florent[1]

[1] Centre d'Ecologie Fonctionnelle et Evolutive CEFE, UMR5175, CNRS, Université de Montpellier, Université Paul-Valéry Montpellier, EPHE, 1919 Route de Mende, 34293 Montpellier Cedex 5, France
[2] Groupe de Spectrométrie Moléculaire et Atmosphérique (GSMA), Université de Reims-Champagne Ardenne, UMR CNRS 7331, Reims, France
[3] Laboratoire des Sciences du Climat et de l'Environnement (LSCE), IPSL, CEA-CNRS-UVSQ, Université Paris-Saclay, 91191 Gif sur Yvette Cedex, France
[4] Institut Méditerranéen de Biodiversité et Ecologie Marine et Continentale (IMBE), Aix-Marseille Université, CNRS, Institut de Recherche pour le Développement (IRD), Avignon Université, 13290 Aix-en-Provence, France

*Correspondence to*: Lilian VALLET (lilian.vallet@cefe.cnrs.fr)

**Abstract.** Fire is considered as an essential climate variable, emitting greenhouse gases in the combustion process. Current global assessments of fire emissions traditionally rely on coarse remotely-sensed burned area data, along with biome-specific combustion completeness and emission factors, to provide near real-time information. However, large uncertainties persist regarding burned areas, biomass affected, and emission factors. Recent increases in resolution have improved previous estimates of burned areas and aboveground biomass, while increasing the information content used to derive emission factors, complemented by airborne sensors deployed in the Tropics. To date, temperate forests, characterized by a lower fire incidence and stricter aerial surveillance restrictions near wildfires, have received less attention. In this study, we leveraged the distinctive fire season of 2022, which impacted Western European temperate forests, to investigate fire emissions monitored by the atmospheric tower network. We examined the role of soil smoldering combustion responsible for higher carbon emissions, locally reported by firefighters but not accounted for in global fire emission budgets. We assessed the $CO/CO_2$ ratio released by major fires in the Mediterranean, Atlantic pine, and Atlantic temperate forests of France. Our findings revealed low Modified Combustion Efficiency (MCE) for the two Atlantic temperate regions, supporting the assumption of heavy smoldering combustion. This type of combustion was associated with specific fire characteristics, such as long-lasting thermal fire signals, and affected ecosystems encompassing needle leaf species, peatlands, and superficial lignite deposits in the soils. Thanks to high-resolution data (approximately 10 meters) on burned areas, tree biomass, peatlands, and soil organic matter, we proposed a revised combustion emission framework consistent with the observed MCEs. Our estimates revealed that 6.15 $MtCO_2$ (± 2.65) were emitted, with belowground stock accounting for 51.75% (± 16.05). Additionally, we calculated a total emission of 1.14 MtCO (± 0.61), with 84.85% (± 3.75) originating from belowground combustion. As a result, the carbon emissions from the 2022 fires in France amounted to 7.95 $MteqCO_2$ (± 3.62). These values exceed by 2-fold the generic GFAS global estimates of 4.18 $MteqCO_2$ (CO and $CO_2$). Fires represent 1.97% (± 0.89) of the country's annual carbon footprint, corresponding to a reduction of 30 % of the forest carbon sink this year. Consequently, we conclude that current European fire emissions estimates should be revised to account for soil combustion in temperate forests. We also recommend the use of atmospheric mixing ratios as an effective monitoring system of prolonged soil fires that have the potential to reignite in the following weeks.

## 1 Introduction

Wildfires recurrently affect European forests, particularly in the southern regions characterized by a Mediterranean climate and northern boreal regions (European Commission. Joint Research Centre., 2023). In contrast, fire activity is significantly



lower in wetter temperate and alpine forests, resulting in relatively less interest and fewer impact assessment studies (Zin et al., 2022). However, this established paradigm of wildfire distribution in Europe may undergo substantial modifications as a result of climate change (Wu et al., 2015). Climate change has the potential to intensify the already recurring fires in the Mediterranean basin under more frequent heat waves (Ruffault et al., 2020) and reshape pyro-regions (Galizia et al., 2023). In

particular, the year 2022 exhibited highly distinctive fire events in the western Mediterranean basin and experienced unusual heat waves and subsequent forest fires in the temperate forests across northern France, Germany, the Czech Republic, and the UK (Rodrigues et al., 2023). These atypical fire events could potentially serve as a preview of future fire distribution, posing a significant risk to temperate forests (Galizia et al., 2023).

However, limited information is currently accessible to assess the impacts of this atypical fire distribution, particularly

concerning carbon emissions into the atmosphere. The gaps in our current understanding of these fires are mainly due to the rare occurrence of such fire distribution within European fire regimes, also impaired by the lack of remote sensing measurements until recently. In a preliminary investigation of fire effects on temperate forests, Vallet et al. (2023) focused on the 2022 fire season as a unique study case. They identified an increased loss of wood biomass in old-growth temperate forests, less affected by fires in the last decades compared to the Mediterranean forests which are mostly affected in their early stage

of forest succession as shrublands. Nevertheless, the impacts of fire on biomass combustion and the resulting carbon emission have not been assessed. Moreover, the combustion of soil, often disregarded in fire-prone Mediterranean ecosystems, remains under-studied due to their thin litter layer and low soil organic content resulting from mild temperatures and high decomposition rates (Jonard et al., 2017; De Vos et al., 2015). The impact of fires on soil carbon stocks is only extensively considered in boreal forests and tropical peatlands where fire incidence is higher (Astiani et al., 2018; Asbjornsen et al., 2005).

However, temperate forests still harbor significant burnable soil carbon pools and peatlands that could contribute significantly to carbon emissions during fires (Muller, 2018; Tanneberger et al., 2017). In these ecosystems, the thick litter layer can be altered by high temperature peaks reached during fire events, and the soil organic layer can propagate fire by the so-called smoldering combustion (Watts and Kobziar, 2013). Smoldering is characterized by a slow, flameless combustion that consumes carbon and releases heat over extensive periods of time. This fire spread mechanism can give rise to overwintering

fires called 'zombie fires', which may reactivate during the subsequent fire season, as observed recently in the boreal region (Irannezhad et al., 2020). Aside from fire safety considerations, these smoldering events could have significant ecological and atmospheric impacts (Watts and Kobziar, 2013) that have been overlooked in impact assessments and in fire emissions from European temperate forests (Van Wees et al., 2022; Wiedinmyer et al., 2023), mostly due to the lack of direct evidences and measurements regarding this process and its extent.

During the year 2022 in southwestern France, the region where the largest managed Pinus pinaster national forest of 'les Landes' stands, firefighters consistently raised concerns about lingering soil fires that posed a potential threat for re-ignition throughout the summer and fall. These fires were eventually expected to dissipate with the arrival of rainfall, which would wash away the burning soil material. However, accurately detecting and monitoring this smoldering combustion using existing Earth Observation Systems has proven to be challenging. Remote sensing methods are less effective in capturing the fire effects



on soils (Johnston et al., 2018) compared to the canopy (Balde et al., 2023; Fernández-Guisuraga et al., 2022) where changes in surface reflectance can be observed due to the biomass combustion during fires (Chuvieco et al., 2019) and due to the energy release detected by thermal sensors (Giglio et al., 2016; Wooster et al., 2021). Unfortunately, the information derived from aboveground assessments of fire emissions does not correlate well with soil carbon losses (Gerrand et al., 2021) due to the complex interactions between plant material and soil properties (Varner et al., 2015). Field observations of fire impacts on

soils are also scarce and mainly focused on boreal peatlands (Turetsky et al., 2011a; Mack et al., 2021) or involve extensive time and effort to assess large-scale areas.

    To fill this research gap on fire impacts on soil stocks and the subsequent carbon emissions across temperate European forests, we leveraged the distinctive extreme 2022 fire season in France as a study case. We hypothesized that the atmospheric signatures of trace gases could serve as a direct indicator of smoldering fires and soil organic matter (SOM) combustion.

Previous investigations of smoldering combustion have shown that this partial combustion results in a high atmospheric $CO/CO_2$ ratio (or inversely correlated to the widely used Modified Combustion Efficiency (MCE) index) in the absence of flaming. Various studies of smoke chemical analysis, including ground-based spectroscopy (Wooster et al., 2011), laboratory burning experiments (Hu et al., 2019), or drone/aircraft campaigns (Lee et al., 2023) have determined MCE indices ranging from 0.6 to 0.8 during smoldering combustion. Recent satellite-based studies based on Sentinel-5P (TROPOMI) retrievals

have confirmed these findings by capturing CO plumes from extreme wildfires (Magro et al., 2021). Notably, Hu and Rein (2022) recently compiled a review on smoldering combustion emission factors, with MCE indices varying from 0.93 for flaming in forests to 0.85 for peatland smoldering combustion. Atmospheric mixing ratios collected by the French monitoring network, part of the Integrated Carbon Observation System (ICOS, 2023) have been used to document MCE indices at the regional scale through its wide continental network of atmospheric towers. Seasonal and interannual variations of greenhouse

gas mixing ratios sampled during extreme climate events have been examined in several studies (Heiskanen et al., 2022; Ramonet et al., 2020). Yet, (Wiggins et al., 2021) remains the only study using the atmospheric tower network to link low MCE values with smoldering combustion to quantify the CO emissions during the 2015 fire season in Alaska.

    In our study, we utilized data from the French atmospheric tower network (ICOS - FR, 2023) collected at stations near the largest fires of 2022 in the temperate forests of les Landes and Brittany, as well as the Mediterranean ecosystems of Provence.

Our objective is twofold : First, to determine if variations in atmospheric MCE could be attributed to fires and to detect smoldering combustion events; and second, to investigate whether regional variations in MCE are related to specific soil and vegetation characteristics, fire spread features, or fire intensity indicated by remotely sensed thermal anomalies. These variables are directly associated with the fire characteristics (Mc Arthur and Cheney 2015), enabling the detection of smoldering combustion. Finally, we utilized our findings to provide an enhanced bottom-up fire carbon emission framework,

benchmarked with the observed MCE indices, and applied it to the 2022 fire season in France. We also compared our emissions to the current global models based on standard fire emission factors (GFAS, 2023) used by the Copernicus Atmosphere Monitoring Service (CAMS | Copernicus, 2023) and publicly delivered in near real-time to stakeholders and society (GFAS | Atmosphere Data Store, 2023). Desservettaz et al. (2022) warned about substantial mismatches among global datasets when



compared to various estimates of fire-induced CO emissions in Australia incorporating surface in situ data, ground-based total column data, and satellite-based measurements. Our study contributes to refining the global greenhouse gas budget for national fire risk assessment, taking into account carbon stocks as an ecological value in the risk assessment framework developed over the European continent (Chuvieco et al., 2023).

## 2 Materials and Methods

### 2.1 Study area

This study focuses on mainland France (41°N-52°N; 5°W-10°E). To facilitate data analysis, we divided the national territory into four regions based on forest communities and fire occurrence (Fig. 1).

- Atlantic temperate forest (Sylvoecoregion A11 to A21 according to the National Forest Inventory (NFI) classification) : This region is primarily characterized by agricultural land, encompassing low vegetation of pasture and cropland. However, this region comprises dense temperate forests hosting deciduous species (*Quercus. petraea*, *Quercus. robur*, *Fagus. sylvatica*, *Alnus. glutinosa*), with a coverage of approximately 11.8%. Historically, this region experienced low fire incidence owing to its humid oceanic climate, with an annual average of 0.013% (±0.006%) of the forest area burned (BDIFF, 2023).

- Atlantic Pine forest (Sylvoecoregion F21 and F22 of the NFI) : This region is almost exclusively covered by extensive maritime pine plantations (Pinus pinaster), cultivated for wood production and covering approximately 76.4% of the region. Although this region experienced a moderate level of fire activity, with an average annual forest burning area of 0.062% (± 0.047%), large fires were reported in 2022 (Vallet et al., 2023).

- Mediterranean forest (Sylvoecoregion J10 to K13 of the NFI): This region is characterized by low, dense forests (covering 39.8% of the region) dominated by species typical of the Mediterranean climate (*Quercus. ilex, Quercus. pubescens, Quercus. suber, Pinus. halepensis*). This region experiences a high frequency of fires, with approximately 0.25% (± 0.21%) of the forest area burned each year.

- Other temperate forests encompass the remaining forested land of France. This region comprises diverse temperate forest communities covering 28.3% of the area, dominated by deciduous or coniferous species and exhibiting varying levels of management intensity. Historically, this region experienced minimal fire occurrence, with an average annual forest burning area of 0.016% (±0.002%)

### 2.2 Fire data

#### 2.2.1 Fine resolution fire polygons

For the fire season 2022, we delimited fire polygons using the semi-automated Burned Area Mapping Tools (BAMTs) (Bastarrika et al., 2014; Roteta et al., 2021). This method was exclusively applied to fires exceeding 30ha and focused on



ignitions spatially and temporally defined with VIIRS data (Schroeder et al., 2014). BAMTS, relying on atmospherically-

corrected and orthorectified images from the L2A product of ESA's Sentinel-2 mission of 2022, involves an algorithm process for deriving three key spectral indices : Normalized Differential Vegetation Index (NDVI) (Rouse et al., 1974), Normalized Burn Ratio (NBR) (Key and Benson, 1999), and NBR2 (García and Caselles, 1991). The VIIRS-derived fire dates facilitated the identification of a pre- and post-burn timeframe to capture the alterations in these three indices, represented using an RGB color scale. Specifically, the pre-fire period extended from the onset of the year (January 1st) up to the date of the fire outbreak

identified by VIIRS. The post-fire period, designed to encompass several weeks beyond the fire outbreak, ensured an adequate number of cloud-free satellite images. Through a visual examination of the RGB spectrum, we manually defined regions as either burned or unburned, which served as training data for a random forest classifier (Belgiu and Drăguţ, 2016). Fine-tuning and quality assessment through visual inspection were performed in Vallet et al. (2023). This key step, unavailable in current automated methods, is required in meeting the international standards advocated by the CEOS Working Group on Calibration

and Validation of remote sensing datasets (Franquesa et al., 2020). Focusing on fires exceeding 30 ha and confined to the fire season (June to September), we identified a total of 70 fire polygons in the year 2022. These fire polygons were primarily located in forested and shrubland areas. Among these fire polygons, three of them located in the proximity of atmospheric towers were chosen for in-depth analysis, referred to as "main fires". These three fires were the largest occurring in each region in the fire season 2022.

## 155 2.2.2 Fire intensity and fire spread

To enhance the precision of our analysis regarding fire behavior during propagation, we incorporated supplementary data, specifically surface thermal anomaly information for active fire detection. This data was gathered from MODIS (Moderate Resolution Imaging Spectroradiometer) instruments on Terra and Aqua satellites (MCD14ML) (Giglio, Louis, 2000), featuring a spatial resolution of 1 km. Additionally, we harnessed VIIRS (Visible Infrared Imaging Radiometer Suite) data

from the SNPP (Suomi National Polar-orbiting Partnership) and NOAA (National Oceanic and Atmospheric Administration) sources, offering a finer spatial resolution of 375 m (Schroeder et al., 2014). The acquisition of these datasets was facilitated through the utilization of the Fire Information for Resource Management System (NASA-FIRMS, 2023). Subsequently, we executed a spatial filtration process to exclude all thermal anomalies occurring outside the confines of our designated fire patches and corresponding to non-forest fires.

The thermal anomalies derived from these data sets were instrumental in our analysis, primarily with respect to assessing the intensity of fires during their propagation phase. We gauged this by examining the Fire Radiative Power (FRP) values, a recognized indicator of combustion intensity (Wooster et al., 2005). Furthermore, to gain insights into the direction and daily rate of fire spread, we leveraged the temporally dated (6-hour intervals) spatial locations of fire hotspots (Fig. 5). Employing an ordinary kriging method, a geostatistical interpolation technique available through the gstat R package (Gräler et al., 2016),

we used the timing (expressed in decimal days) as the target variable for interpolation, similar to previous studies (Parks, 2014; Veraverbeke et al., 2014; Scaduto et al., 2020). For each main fire, we manually fine-tuned a Gaussian or Spherical function





to derive the best-fitted variogram. The result of this fire spread mapping is exemplified in Fig. 5. Finally, we computed the hotspot density (number per hectare) within each fire polygon over the entire fire duration. This approach allows us to capture protracted soil and peatland fires that exhibit either a heightened hotspot density or an extended burning period (Usman et al.,
175  2015).

### 2.3 Atmospheric CO/CO$_2$ mixing ratio analysis

In this study, we collected hourly measurements of CO and CO$_2$ mixing ratios derived from a subset of instrumented towers part of the French monitoring network (SIFA, 2023), a network established for monitoring atmospheric greenhouse gas variations in the atmosphere. These measurements were conducted with high-precision cavity ring-down spectroscopy
(CRDS), with up to three sampling levels (Conil et al., 2019; Lelandais et al., 2022; Lopez et al., 2015; Schmidt et al., 2014). The selected stations, outlined in Table 1, include distant stations and nearby stations located within 20 km of the 2022 large fires that occurred in the Atlantic temperate forests (Brittany), Atlantic pine forests (Landes), and Mediterranean forests. Data collection for this study spanned from June 15th to September 1st, 2022. In the context of the Atlantic pine forest, the dominant winds were from the northeast, propelling the plume seaward. Notably, a shift in wind direction occurred on July 14th-15th,
with the wind veering to the north-northwest. This shift contributed to the highest CO peaks observed at the Biscarrosse (BIS) station. Subsequently, on the 19th, the wind shifted westward, transporting the plume inland and leading to elevated CO concentrations at distant stations. Similarly, in the Atlantic temperate forest (Brittany), predominant winds came from the northeast, steering the plume away from the Roc'h Trédudon (ROC) station toward the ocean. Changes in the wind direction led to intermittent CO signals at the ROC station. The only instance when the plume was transported inland occurred on July
19th.

To determine the locations of the sources corresponding to the identified CO mixing ratio anomalies observed at the atmospheric towers, we computed back-trajectories representing the different air masses sampled at the tower locations. This step was accomplished using the Hybrid Single Particle Lagrangian Integrated Trajectory (Hysplit) model (Stein et al., 2015). In a backward-in-time configuration, particles were released from the receptor site and monitored over 7-day intervals. The
Global Forecast System (GFS) meteorological model (National Centers For Environmental Prediction/National Weather Service/NOAA/U.S. Department Of Commerce, 2015) provided the atmospheric conditions (wind and turbulence) to drive these particles from the receptors to the sources in the Hysplit simulations. The GFS outputs, featuring a horizontal resolution of 0.25° x 0.25° and 3-hourly time intervals, served as the meteorological inputs. We conducted Hysplit simulations in a forward-in-time configuration releasing particles (600 per hour) from the fire locations, over the fire duration from the exact
burned area. By tracking the arrival times of these particles within an influence region surrounding each atmospherictower, we successfully attributed a source to each anomaly. These influence areas featured varying radii to account for transport uncertainties, considering that the minimum distance between the towers and the nearest fires ranged from 7 to 650 km. For towers in proximity to active fires (within 20 km), the influence radius was set at 4.5 km, corresponding to a single grid cell.





For more distant towers, the influence radius was extended to 25 km to account for errors associated with long-distance transport.

To quantify the excess in CO and $CO_2$ mixing ratios originating from the fires, we needed to determine the background concentration levels that would have been observed in the absence of fires. Due to the extensive duration of some observed fire events (>10 hours), a simple interpolation method could not be used without impacting our enhancements with variations in the background air (diurnal cycle, sea breeze periods…). To determine the background flow more accurately, we trained a Random Forest (RF) regression model for each gas at each station. The RF model is a non-parametric statistical method based on averaging over ensembles of multiple regression trees (Breiman, 2001). In our approach, we randomly divided the atmospheric observations into three categories: 1) the studied data, 2) the training data, and 3) the testing data. Initially, we isolated the data that were indicative of forest fires contributions to the observations. These periods were characterized by elevated CO mixing ratios and were automatically identified as outliers by the Tukey's fence approach (Tukey, 1977). Subsequent manual quality checks ensured that the flagged data coincided with the active forest fire periods. The remaining data were then divided into training (70% or approximately 1000 data points) and testing (30% or around 400 data points) sets for each station separately individually. In addition to the mixing ratios, meteorological and calendar data were included as input variables for the RF models. The meteorological data encompassed parameters such as 10 m wind speed and direction (m.s-1), 2 m Temperature (°C), and Boundary Layer Height (BLH) (m). These meteorological parameters were extracted from the ERA5 hourly reanalysis dataset (Hersbach et al., 2020). Time-derived variables included the hour of the day, day of the week, day of the month, and month of the year. For the RF model, the number of regression trees was set at 100.

The RF model performance was assessed using the testing data, with evaluation metrics including the correlation coefficient (R) and the root-mean-square error (RMSE). The model's performance scores exhibited variability across sites. On average, we achieved a correlation of 0.883 and 0.98, along with an RMSE of 7.66 ppb and 1.12 ppm for CO and $CO_2$, respectively (Table 1).

The excess mixing ratios of CO and $CO_2$ attributable to the fires, denoted as $\Delta[CO]$ and $\Delta[CO_2]$, were calculated as the difference between the observed mixing ratios and the simulated background mixing ratios generated by our RF model. Subsequently, we computed the modified combustion efficiency (MCE), with values indicating higher levels during flaming fires combustion and lower levels during smoldering fires, according to Equation (1) (Hao and Ward, 1993; Yokelson et al., 1996):

$$MCE = \frac{\Delta[CO_2]}{\Delta[CO_2] + \Delta[CO]}$$

(1)

**2.4 Above- and below- ground dry matter stock**

To further comprehend the origin of the MCE observed at the monitoring towers, we sought to estimate the pools affected by the fires, possibly contributing to the emissions of CO and $CO_2$. Given that our analytical framework relies on emission factors (EF) expressed in grams of gas emitted per kilogram of dry matter (DM) consumed, we expressed these pools in units of tons





of dry matter. The entirety of the ecosystem dry matter stock is partitioned into two distinct types : the aboveground stock (AGS) and the belowground stock (BGS). Each of these stock types encompasses multiple pools. The AGS comprises the stem, branch, leaf, shrub, grass, and litter pools, while the BGS includes Soil Organic matter (SOM), peat, and lignite pools.

### 2.4.1 Forest stem and branch pool

Within the AGS affected by fires, the stem and branch pools are prominent components. These pools align with the woody AGB-L (Above-ground biomass loss) method introduced by Vallet et al. (2023). This method is based on two high-resolution data sources: first, a 10-m resolution mapping of vegetation height obtained from GEDI, Sentinel 1, and 2 satellite images from 2020 (Schwartz et al., 2023); and second, data indicative of forest communities and individual descriptors, sourced from French National Forest Inventory (NFI) since 2005 (IFN, 2023a). Data supplied by the NFI within a 5-km radius of fire was used to delineate individual and population allometric relationships.

Based on the remotely-sensed data on vegetation height, we estimated the biomass of a model tree within each burned pixel. Subsequently, for each pixel, we determined a tree density based on the biomass of the model tree and the density-dependency relationship derived from NFI data. After applying the AGB-L method to each 10-m burnt pixel, we segregated the above-ground forest biomass into stem pool and branch pools. Deciduous branches accounted for 39% of the above-ground biomass, while coniferous branches contributed 25% (Loustau, 2010).

### 2.4.2 Shrub, grass, and litter pools

To account for AGS affected on non-forest pixels (where the height is less than 3m), we applied a fixed biomass (dry weight) density value of 10tDM.ha$^{-1}$ for shrubland vegetation and 4tDM.ha-1 for herbaceous vegetation (Vallet et al., 2023). These values are in agreement with the stocks included in the FINN carbon emission model (Wiedinmyer et al., 2023). Pixels were classified as containing shrubland vegetation based on the presence of sclerophyllous vegetation in the CORINE LAND COVER database (CORINE Land Cover 2018, 2023), along with a recorded vegetation height below 3m. Pixels not classified as forest or shrubland were considered as grassland.

The litter pool was also incorporated into the AGS. It was derived from the GFED5 dataset, available at a resolution of 500-m by (Van Wees et al., 2022). We resampled this fine litter data to a 10-m resolution using the nearest-neighbor method.

### 2.4.3 Forest and shrubland leaf pool

The leaf pool, representing the fraction of vegetation most completely consumed during combustion, was quantified based on a combination of satellite data and in situ measurements of leaf traits. Leaf area index (LAI) data at a resolution of 300m were derived from the Sentinel-3 LAI product provided by the Copernicus service (Verger et al., 2014). These data were compiled over the summer period of 2022 (June to September), and the average of the non-zero values for each pixel was extracted. Specific Leaf Area (SLA, in m2.kgDM-1) was obtained at a resolution of 500 m from the TRY database (Moreno-Martínez et al., 2018). To calculate leaf mass, we initially conducted a nearest-neighbor resampling of LAI and SLA maps at 10 m





resolution. Subsequently, the leaf pool density (kgDM.m$^{-2}$) was determined by dividing the LAI values (m$^2$.m$^{-2}$) by the SLA values (m$^2$.kgDM$^{-1}$) for each pixel. Only pixels categorized as forest or shrubland (height >3m) were included in this leaf pool

dataset.

Consequently, the AGS is then composed of 6 pools : stem, branch, leaf, shrub, grass, and litter.

### 2.4.4 Soil Organic Matter (SOM) pool

The Soil Organic Matter (SOM) is encompassed within the BGS. Data for this pool was sourced from the European Soil Data Centre (ESDAC) (yigini & panagos, 2016), offering carbon density values (tC.ha-1) for the top 20 cm of soil at a resolution

of 1000 m. To determine the pool of soil organic matter within each burned pixel, we converted these carbon values into organic matter, assuming a carbon content of 0.5 (Pribyl, 2010). This data was then resampled at 10-m resolution using the nearest-neighbor approach.

### 2.4.5 Other belowground pools : peatland and lignite

In order to investigate the sources of smoldering combustion and pyrolysis, we considered two additional pools within the

BGS. Marshland areas, particularly peatland, can potentially contain huge amounts of organic matter, which is often assumed as insignificant in temperate forest fire emissions. During the summer, waterlogged areas can become vulnerable to fire as they dry out. To account for peatland areas, we relied on the CORINE LAND COVER (CLC) database (CORINE Land Cover 2018, 2023). We established a fixed characterization of the peatland, assuming a depth of 2 m and a mass density of 145 kgDM.m-3, as measured in France (Pilloix, 2019). We then calculated the pool mass for any point within the CLC polygon by

multiplying the pixel area (~100 m²) by the depth and biomass density.

Lignite is a distinctive pool within the BGS found in 'Les Landes', arising from a slow decomposition process. Historically, lignite has been utilized as an energy source in Les Landes, near the city of Hostens, for its high concentration of carbon. Firefighters in this area reported high soil temperatures near the ancient mines. The lignite layer is near the surface and located beneath the organic soil. The location of the lignite area was provided by the APPHIM association (apphim.fr - Les gisements

de charbon et lignite, 2023) around the Hostens village. The lignite mine typically has a depth ranging from 2 to 5m, extending to 10-15 m. For our analysis, we assumed a fixed depth of 2 m (Le lignite d'Hostens, 2023). The bulk density of brown coal generally hovers around 700kgDM.m-3 (Coal - Carbon, Organic Matter, Sedimentary Rock | Britannica, 2023). Accordingly, the density of the lignite pool was set at 1400kgDM.m-² of burned surface. This particular pool of carbon has been affected by two large fires during the 2022 fire season.

Thus, the BGS encompasses three pools: Soil Organic Matter (SOM), peat, and lignite.

### 2.5 Carbon emissions

Utilizing information from fire polygons (Fig. 2, 'Database') and estimation of AGS and BGS pools (Fig. 2, 'Stock'), we quantified $CO_2$ and CO emissions arising from two combustion phases, namely, flaming (F) and smoldering (S). This



quantification was computed for each of the AGS (stem, branch, leaf, shrub, grass, litter) and BGS (SOM, peat, lignite) pools.

Emission assessment was facilitated by accounting for two crucial factors : the combustion completeness (CC), denoting the proportion of pool altered by combustion, and emission factors (EF, in g.kg-1DM) for $CO_2$ and CO. For each individual pixel within the fire patch ($p$), each specific pool ($P$) (Table 2) and each gas ($x$), we calculated emission ($E$) using the following formula (2) :

$$E_{Px} = M_P * CC_P * (SF_P * EF_{Pxs} + (1 - SF_P) * EF_{Pf}) \qquad 305 \qquad (2)$$

$E_{Px}$ : Emission of gas $x$ from pool $P$ (g)
$M_P$: dry Mass of pool $P$ (kgDM)
$CC_P$ : Combustion completeness of pool $P$ (percentage of available pool)
$SF_P$ : Smoldering fraction of pool $P$ (percentage of combusted pool in smoldering phase)
$EF_{Pxs}$ and $EF_{Pxf}$ : Emission factors for pool P into gas $x$, during smoldering ($s$) and flaming ($f$) phase. (g.kg-1DM)

To calculate the emissions of gas $x$ (Fig. 2, 'Emission') from all pools ($n$ pools $P$) within each burned pixel ($p$), we utilized the following equation (3) :
$$E_{px} = \sum_{P=1}^{n} E_{Px} \qquad 315 \qquad (3)$$

Consequently, we were able to obtain an aggregated emission value for gas $x$ encompassing the entire fire ($F$) comprising $m$ individual pixels $p$, as specified in equation (4) :

$$E_{Fx} = \sum_{p=1}^{m} E_{px} \qquad (4)$$

Table 2 provides a comprehensive summary of CC, EF, and SF for each pool, drawing from a bibliographical review of available data from global fire emission models, such as GFED (Van Wees et al., 2022) and FINN (Wiedinmyer et al., 2023),
along with empirical field measurements conducted in temperate forests. Notably, in the absence of specific data synthesis for Europe, the fraction of smoldering combustion for each pool was inferred from data collected in American temperate forests (Prichard et al., 2020).

To establish a comparative baseline between our fire-level total emissions and the hourly MCEs derived from measurement obtained by the atmospheric towers, accounting for the temporal dynamics of fire spread, we delineated three distinctive phases
in the propagation of each fire :

1) The flaming phase (FP), where the AGS constitutes the entire combustion. 50% of AGS is affected during this phase.

2) The mixed phase (MP), characterized by ongoing aboveground flaming at the fire front while smoldering combustion consumes the wood residual and BGS over the previously burned area. This phase involves 50 % of AGS and 25% of BGS.

3) The smoldering phase (SP), devoid of flaming but marked by continuing smoldering in the soil and wood residuals,
representing the totality of emissions. 75 % of BGS is impacted during the smoldering phase.

As a point of reference for comparison, we utilized the Global Fire Assimilation System (GFAS, 2023) dataset for fire emissions (Kaiser et al., 2012). This dataset is the only to offer near-real-time coverage extending up to 2022, generating daily emissions based on MODIS MCD thermal 'hotspots' anomalies and biome-specific standard emission factors (in kgDM.MJ-1). GFAS delivers information at a 0.1° resolution, covering burnt dry matter, fire emissions, and injection height on a daily



basis since 2003, with near-real-time updates. We accessed GFAS data for $CO_2$ and CO emissions for the period spanning

from June to September 2022, considering the entire dataset within this timeframe for our analysis.

## 3 Results

### 3.1 Attribution of the MCE to the various fires

In order to disentangle the inherent CO and $CO_2$ background mixing ratios at the atmospheric tower stemming from prevailing

atmospheric conditions, and the emissions originating from actual fires, we initiated a rigorous assessment of our Hysplit

atmospheric transport simulations and their alignment with the detected tower overpasses. Fire plume shapes and directions

can be qualitatively evaluated when smoke is visible in visible satellite imagery. Figure 3 visually demonstrates the

correspondence between observed plume positions, detected by MODIS, and the modeled plume positions, particularly in the

case of the Landes fires. Notably, both the observed and modeled plumes exhibited a correct overlap, reinforcing the precision

of our modeled wind direction changes as corroborated by the analysis of the comprehensive suite of satellite snapshots

available throughout the study period.

It is worth mentioning that, during the same study period, TROPOMI data showed the arrival of an air mass with elevated CO

concentrations from Spain, where forest fires were occurring at the same time (not shown here). However, we did not account

for those fires in the current study, since the analysis of the HYSPLIT Lagrangian model results indicated a minimal impact

from these fires on the time series monitored at the French towers, as evidenced by both forward and backward-in-time

simulations. Specifically, the results of the Lagrangian model showed that the stations CRA and PUY were largely unaffected

by these fires. The plumes from both the Landiras and Mont d'Arrée fires were mixed before reaching the inland stations of

MDH, OPE, SAC, TRN. Consequently, we opted to exclude these towers from the MCE analysis, reserving their data solely

for the evaluation of the RF background estimates. At each of the three remaining sites, namely BIS, OHP, and ROC, only the

influence of the adjacent fire was observed: Landiras1 for BIS, La Montagnette for OHP, and Monts d'Arrée for ROC.

The analysis of the MCE index during the days when the simulated particles reached the atmospheric tower locations shows

that the MCE signatures associated with the fires exhibit regional variations. In particular, the fire near BIS displayed an

average MCE of $0.83 \pm 0.03$, the lowest mean value among the three sites (Fig. 4). The BIS site shows mostly low minimum

values, observed most often under smoldering combustion phases and high-temperature pyrolysis phases. In contrast, the OHP

fire predominantly featured MCEs exceeding 0.95, marked by low variations, with a minimum value of 0.93, primarily

observed during flaming combustion. The ROC site collected intermediate values , with a mean MCE of 0.94, close to the

Mediterranean MCE observed at OHP. However, ROC exhibited minimum values that reached 0.82, far beyond the values

observed at OHP. This variation suggests the occurrence of smoldering combustion phases throughout the fire propagation.

Daily MCE variations (Fig. 4) emphasized a decreasing trend for the BIS fire, indicating an increase in smoldering combustion



over time, supporting the hypothesis of a prolonged soil combustion following the cease of flaming phase. Conversely, this temporal pattern was less discernible for the fast-spreading ROC fire.

Furthermore, we looked into the 1-minute averaged concentrations to investigate rapid changes in combustion, fire propagation, atmospheric transport, and the implications of different averaging periods on our analytical results. We found that the MCE values derived from both the 1-minute and 1-hour averaged mixing ratios are consistent, as shown in Fig. 4.

While there is a broader dispersion in the case of the 1-minute sampled mixing ratios, the fire MCE signal remained consistent across all stations. Notably, when accounting for the uncertainty in the RF estimates, the MCE varied by 2% when propagating the mean error from the RF model for CO and $CO_2$. This variation had no discernible impact on the overall findings of this study, ensuring the consistent differentiation of the combustion types attributed to the main fires.

### 3.2 Exposure and stock affected

To disentangle the fire behaviors associated with the observed MCE indices measured at the towers located within the Atlantic temperate forest (ROC), Atlantic pine forest (BIS), and Mediterranean forest (OHP), we performed a comprehensive characterization of the affected AGS and BGS by these main fires.

The ROC fire, encompassing a total area of 1,726 hectares, primarily impacted low vegetation, with grassland covering 63.3% of the burned area (Table 3 and Fig. A1). The fire's influence on forest area was comparatively limited, spanning only 129 ha,

characterized by a low biomass density of approximately 46tDM.ha$^{-1}$. A distinguishing feature of this fire is the substantial presence of peatland, occupying 449ha (26% of the burned area). Remarkably, the aggregated stock, combining AGS and BGS, is largely dominated by the peatland pool, accounting for 86.9% of the total stock. We note here that this pool is recognized for its propensity to combust predominantly through smoldering.

The BIS fires extended over a considerably larger area of 12,140 hectares and predominantly affected forested areas (71% of

the burned area) characterized by high biomass density ranging from 20 tDM.ha-1 to 150 tDM.ha$^{-1}$ (see Fig. A1 'Vegetation'). Moreover, the SOM in this region falls within the highest range of the country, varying between 210 and 250 tDM.ha$^{-1}$, a noticeably larger amount compared to the temperate Atlantic (100-220 tDM.ha$^{-1}$) and Mediterranean (70-120 tDM.ha$^{-1}$) regions (Fig A1, 'SOM'). Additionally, this fire also altered 61 hectares of peatland. An unusual feature of this area is the presence of a lignite layer situated near the surface, spanning 1,909 hectares within the burned area (15.7%). Remarkably, the

lignite pool constitutes 88.0% of the total dry matter stock (AGS and BGS), followed by the SOM pool (9.4%). These two significant pools, lignite (combusted at high temperature during the pyrolysis phase) and SOM (mostly smoldering), both contribute to a substantial stock of carbon that is potentially affected, resulting in low MCEs.

Finally, the OHP fire in the Mediterranean region primarily affected forests (76.1%), along with low vegetation zones like garrigue (shrubland = 15.3% and grassland = 8.6%). Forest biomass in this area, however, falls within the low range of biomass

density observed in the country, with a median of 60.4 tDM.ha$^{-1}$, and the soil contains relatively low amounts of organic matter (95.2 tDM.ha$^{-1}$). Conversely, the aggregated stock (BGS and AGS) density, amounting to 147 tDM.ha$^{-1}$, stands in stark contrast to the fires in Atlantic pine forests (2,502 tDM.ha$^{-1}$) or Atlantic temperate forests (867 tDM.ha$^{-1}$).



As a first step toward identifying potential factors contributing to the lower MCEs in the BIS and ROC fires, we illustrate here that the fires with the lowest minimal MCEs (ROC, BIS) occurred in areas marked by the highest belowground organic density. Smoldering features shown by these fires have been either favored by carbon-enriched zones, such as peat bogs or lignite, or, as seen in the Landes region, featured a high SOM density.

### 3.3 Fire characterization

To discern whether specific fire characteristics could effectively distinguish fires affecting BGS, we conducted an assessment based on key parameters, such as the extent, duration, rate of spread, and intensity with 6-hourly Fire Radiative Power (FRP). Among the study sites, the maximum FRP was observed during the OHP fire, reaching 359 MW, followed by BIS with 299 MW and ROC with 150 MW (Fig. A2). ROC and OHP fires exhibited a relatively short duration of high FRPs, extending up to three days, in contrast with the BIS fire, where the period of high FRP persisted for eight days. However, when examining low-intensity FRPs, a discerning pattern emerged. The OHP fire showed no remaining burning activity beyond the initial three days of high-intensity combustion. In contrast, the ROC and BIS fires exhibited a protracted signal, spanning up to 25 days after ignition for ROC and 32 days after ignition for BIS (Fig. A2). This information appears pivotal for distinguishing fires characterized by low MCEs.

Furthermore, an evaluation of the fire rate of spread (ROS) within the burned area (Fig. 5) revealed distinct patterns. The BIS fire displayed a notably high hotspot density of 0.27 hotspot.ha$^{-1}$, combined with a relatively slow ROS at 0.147 km.h$^{-1}$. In contrast, the ROC fire expanded rapidly (median ROS = 1.77 km.h$^{-1}$), along with a markedly lower hotspot density of 0.055 hotspot.ha$^{-1}$. In particular, this fire spread relatively rapidly over grasslands, even when compared to the OHP fire, which occurred over shrublands and Mediterranean vegetation (0.66 km.h$^{-1}$ with 0.05hotspots.ha$^{-1}$).

Based on the characteristics related to propagation and combustion, we conclude that fires prone to experiencing smoldering combustion, such as BIS and ROC fires, exhibit a prolonged duration of hotspots after ignition, which is not observed for the OHP fire. Interestingly, the median ROS or maximum fire intensity does not appear to be discriminating factors between fires impacting aboveground and belowground stocks.

### 3.4 Bottom-up approach on carbon emissions

Leveraging our estimation of both AGS and BGS in each of BIS, ROC, and OHP fires, we undertook a bottom-up assessment of MCEs. This assessment compared our MCE estimates to the ranges of combustion and emission factors values estimated by previous studies. In our initial approach, we conducted the basic calculations akin to those employed in global fire emissions models for temperate forests, exemplified by GFAS and FINN). This approach exclusively accounted for AGS and focused only on flaming combustion (Table 4, 'AGS only'). The resulting MCEs ranged from 0.955 to 0.961 for all the fires, with no significant distinctions between them. While these values closely mirrored the MCEs observed at the OHP tower, they notably diverged from the MCEs captured at the ROC and BIS stations.





In our subsequent approach, we incorporated belowground combustion effects for ROC and BIS. We divided the combustion
process into three distinct phases (flaming phase, mixed phase and smoldering phase). For the ROC fire, the calculated MCE
values for the flaming phase were 0.961 (± 0.001), aligning with the median value obtained from the hourly mixing ratios
measured at the ROC tower. Subsequently, for the mixed phase, MCE values of 0.828 (± 0.015) were derived, corresponding
to the lower range of 1-h mixing ratios. Finally, for the smoldering phase, MCE values of 0.796 (± 0.001) were obtained,
similar to the minimum values observed within the distribution of the 1-min mixing ratio.

Considering the BIS fire, the results for the flaming phase exhibited MCE values of 0.956 (± 0.004), values corresponding to
the upper bounds of observations collected at the BIS tower. Subsequently, for the mixed phase, MCE values of 0.821 (±
0.015) were calculated, representing the respective median values from the 1-hour mixing ratio and the 1-min MCE. Finally,
for the smoldering phase, an MCE of 0.729 (± 0.011) was derived, indicating a significant occurrence of smoldering
combustion rate, and closely mirroring the minimal values obtained for the 1-hour MCE measured at this tower.

This refined bottom-up approach, including soil smoldering combustion, successfully captured the spectrum of MCEs observed
at the ICOS atmospheric towers. These findings, which could not be obtained from aboveground combustion alone, underscore
the significance of accounting for belowground combustion when addressing the carbon emission budget.

## 3.5 Fire emissions assessment in 2022 for France

Drawing from our MCE-derived carbon emissions estimates of AGS-BGS combustion, we applied our refined carbon emission
framework to the 70 fires exceeding 30 ha, which were accurately mapped across France. Smoldering combustion was
exclusively attributed to fires affecting vegetation types similar to the BIS and ROC fires, namely those encompassing needle
leaves, peatlands, and lignite.

The year 2022 witnessed a significant impact of fires in the Atlantic pine forest region, with a total burned area of 26,850 ha
(Fig. 6), constituting 64.5% of the overall burned area. Ranked second, the Mediterranean region experienced several fires
over 7,600 ha, accounting for 18.2% of the total burned area. Fires mainly altered forest areas in the Atlantic pine region
(76.5%) and other forest (75.6%) regions. Regarding the Mediterranean region, fires influenced both forest (45.4%) and low
vegetation, including shrubland (11.0%) and grassland (43.6%). In the Atlantic temperate forest, grasslands were the most
affected, encompassing 59.2% of the burned area.

In our estimation, out of the total 44.68 MtDM of stock impacted by fires in 2022 and potentially lost, only 4.526 (± 2.138)
MtDM was actually combusted and directly released into the atmosphere (Table A1). The Atlantic pine forest region
contributed to the majority of this combusted matter due to its particularly high burned area and its substantial densities of
AGS and BGS. More precisely, its AGS accounts for 28.2% (± 1.9), and its BGS for 54.1% (± 2.6). Moreover, the Atlantic
temperate forest contributed significantly to the total stock combusted, when considering BGS, primarily due to the presence
of peatlands, accounting for 5.2% ± 0.3. In contrast, AGS combustion in the other three regions outside the Atlantic pine forest
was responsible for only 12.5% (± 0.9) of the total stock loss.



Our estimates indicate that the fires of 2022 directly emitted 6.154 ($\pm$ 2.650) Mt of $CO_2$, with AGS and BGS contributing nearly equally to these $CO_2$ emissions. Specifically, all AGS were found responsible for 49.5 ($\pm$ 2.9) % of the annual $CO_2$ emissions, with the remainder attributed to BGS, particularly SOM and lignite from the Atlantic pine forest region (46.4 $\pm$ 2.7%). In comparison, the GFAS framework estimated that summer fires were accountable for 3.86 $MtCO_2$ emissions when

excluding belowground combustion, a value that corresponds to the lower bound of our estimations.

Taking into account soil combustion, we reach a value of 1.147 ($\pm$ 0.615) MtCO emitted into the atmosphere. BGS combustion dominates the total CO emissions, representing 87.3 ($\pm$ 0.8) % of the annual emissions. We also note that the Atlantic pine forest region, through the combustion of its SOM and lignite, accounted for 81.6 ($\pm$ 0.6) % of the CO emissions. In stark contrast, GFAS provided markedly lower CO emissions with 0.204 MtCO emitted during the 2022 fire season, which is 5.6

times lower than our estimates when excluding belowground combustion.

## 4 Discussion

### 4.1 Remote sensing fire characterization for carbon emissions : beyond burned area

Remote sensing information has played a key role in advancing our understanding of fire characteristics and their effects. Various studies have employed remote sensing data to examine various aspects such as estimates of burned areas (Chuvieco

et al., 2019), fire sizes derived from aggregating burned pixel (Andela et al., 2019; Artés et al., 2019; Laurent et al., 2018, 2019), fire spreading patterns based on burn dates within fire patches (Benali et al., 2016; Chen et al., 2022; Cardíl et al., 2023), fire intensities determined by fire radiative power (Wooster et al., 2021), and fire severity assessment (Alonso-González and Fernández-García, 2021). While these advancements provide valuable insights to characterize key features of fires driving combustion and carbon emission processes, it is important to acknowledge their limitations. These include the difficulty in

detecting small fires, which can lead to an underestimation of burned areas (cf. Mouillot et al., 2014 for review), as well as challenges in accurately assessing fire intensity (Freeborn et al., 2014). Additionally, uncertainties persist in detecting burned areas in the forest understorey (Roy et al., 2006), as well as in soils, peatlands (Atwood et al., 2016) and croplands (Hall et al., 2021). Combining information from both soil vegetation fire types (Fisher et al., 2020; Sirin and Medvedeva, 2022) also remains a complex task. Efforts are currently underway to address these limitations through the development of more refined

methods. These improvements encompass obtaining finer resolution data for burned area (Chuvieco et al., 2022), enhancing the detection of understorey fires (East et al., 2023), and providing more frequent and higher-resolution FRP datasets, such as those from VIIRS or stationary FRP information (Mota and Wooster, 2018). The use of hyperspectral sensors is also anticipated to offer new opportunities for improved fuel mapping, fire severity assessment and combustion analysis (Veraverbeke et al., 2018).

Based on current remote sensing strengths and weaknesses in fire characterization, we employed here the most detailed available data on burned areas and aboveground biomass in France. This fine-resolution dataset shows significant differences in burned estimates when compared to coarser resolution information (Vallet et al., 2023). We augmented this dataset with





additional information on fire intensity, duration and ROS, all of which were calculated from 6-hourly VIIRS FRP data, as has been done in previous studies in different regions (Benali et al., 2016; Chen et al., 2022; Cardíl et al., 2023).

An interesting addition to our analysis was the estimation of fire ROS, which exhibited considerable variability. ROS ranged from 1.7 km.h-1 in Brittany, predominantly affecting heathlands, to 0.7 km.h-1 in the Mediterranean basin, and even reached a significantly lower level in les Landes not exceeding 0.2 km/h. Our estimates of fire spread fall within the range of previous ROS estimates, which have varied from 0 and 30 km.day-1 (equivalent to 0-1.25 km.h-1) in California (Hantson et al., 2022), with notable impacts observed when ROS exceeds 0.8 km.day-1 and intensity surpasses 0.8MW. For instance, Cardíl et al.

(2023) estimated ROS values of 0.12, 0.17, and 0.19 km.h-1, respectively for heathland, broadleaves, and pine forest based on hotspot data, while Salis et al. (2016) utilized fire spread models to estimate ROS ranging from 0.12 to 3.6 km.h-1. However, higher ROS have been observed in grasslands, ranging from 1.6 to 17 km.h-1 (Cruz et al., 2022). Mediterranean fires are known to be predominantly wind-driven in southern France (Ruffault and Mouillot, 2015), resulting in fast and unidirectional fire spread patterns, which limits long fire residence time affecting soils. The northern region of France is windy on the Britany

coast and northern Channel shores, but wind speed remains lower across the southwest (Landes). Additionally, the Atlantic influence of fast-moving low-pressure systems going from West to East leads to daily changes in wind directions, as opposed to the long-lasting unidirectional Mistral winds along the Mediterranean coast (Soukissian and Sotiriou, 2022). A noteworthy aspect related to intensity $I$ (in MJ) is its relationship with heat release $H$, fuel consumption $w$, and rate of spread $R$ (Alexander and Cruz, 2012). For a given intensity and heat release, fuel consumption is inversely related to ROS due to increasing

residence times. This relationship suggests that slower fires may be more prone to consume larger fuel loads (Cobian-Iñiguez et al., 2022).

Regarding peatlands, previous studies have reported varying ROS values, with Cardíl et al. (2023) referring to 0.12 km.h-1 based on remotely sensed hotspots, while Huang and Rein (2017) only report 10 cm.h-1. This indicates that hotspots over peatland might represent the flaming of the surface, whereas the actual combustion of peat and fire progression occurs at a

much slower pace and with lower intensity, making it challenging to fully capture by thermal anomalies.

In summary, our exploration of fire spread processes in France has shwon that the duration of hotspots within fire patches could serve as an effective and near-real-time indicator of soil combustion, which is closely related to smoldering combustion, and, in turn, toshown the low MCE values. This information on hotspot duration within fire patches has the potential to provide early warning signals for both populations and stakeholders, alerting them to potential air quality issues and the possibility of

reignition (Xifré-Salvadó et al., 2020). Additionally, we recommend including this information as an additional key variable describing fire events in global fire patches databases (Laurent et al., 2018).

### 4.2 Pre-fire carbon stocks uncertainties

In addition to assessing the extent of burned areas, the accuracy of carbon emissions estimates is contingent upon the precision of the available biomass available for combustion. Recent enhancements in tree density and biomass estimation, encompassing

isolated trees (Brandt et al., 2020) and more refined tree height data from Lidar (Schwartz et al., 2023), have played a crucial





role in improving the reliability of such estimates. These advancements, which we incorporated into our methodology, have been discussed in Vallet et al. (2023).

Estimates of SOM at regional and global levels (Lin et al., 2022; Vanguelova et al., 2016) have historically exhibited a relatively large level of uncertainty. We decided to rely on the ESDAC database (Yigini and Panagos, 2016), a strategy

consistent with SOM observations available across the country (Martin et al., 2019). It is worth noting that deeper soil conditions better correspond to soil carbon information derived from biogeochemical models (Van Der Werf et al., 2017; Van Wees et al., 2022).

Exploring the effects of fires on the depth of soil burning has been a relatively understudied domain at a large scale. There is potential for improvements through Lidar technology, which enables the identification of changes in soil surface thickness

resulting from combustion (Reddy et al., 2015; Mickler et al., 2017), including low-severity peat fires (Bourgeau-Chavez et al., 2020). Peatlands, with their substantial stores of SOM, are susceptible to vertical spread rates, estimated at around 1 cm.h$^{-1}$ by Huang and Rein (2017) , or approximately 0.8 cm.h$^{-1}$ (0.-2.3 cm.h$^{-1}$) in tropical peatlands (Graham et al., 2022). To maintain a conservative approach, we adopted a ROS of 0.2 cm.h$^{-1}$ for soil combustion, resulting in a daily consumption of approximately 4.8 cm, which roughly corresponds to 40 cm burned over an 8-day period, which corresponds to the average

flaming duration of our fires. This 40 cm of consumed peat aligns with the upper bound of our soil combustion parameters, while conventional peatland emissions models often assume 20 to 30 cm of peat being burned (Kohlenberg et al., 2018). However, it is worth noting that these parameters can vary from 1 cm to 54 cm in temperate peatlands in the UK (Davies et al., 2013). With these parameters, we reached an estimate of carbon emission of 172 ($\pm$ 74) tC.ha$^{-1}$ emitted, which is slightly higher than the value of 96tC/ha estimated by Davies et al. (2013) for US temperate forests. For a comparative perspective,

Mickler et al. (2017) using fine resolution LIDAR data revealed that peatland wildfires could exhibit an average burn depth of 42 cm, resulting in an average belowground carbon emissions estimated at 544.43 t C ha$^{-1}$. In terms of peatlands in France, the Corine Land Cover (CORINE Land Cover 2018, 2023) was utilized to identify their exposure to fires. According to this source, the extent of wetland (marshland and peatland) in France stands at around 89,000 ha. However, we note here that this information remains highly uncertain, with different estimates varying between 275,000 ha and 300,000 ha according to

Tanneberger et al. (2017). This peatland extent would represent 0.52% of the country, out of which, 75,000 to 100,000 ha are considered as mires. For another comparison point, Muller (2018)estimated the extent of french peatland at 59,000ha.

## 4.3 Atmospheric assessments of combustion

In addition to bottom/up approaches that rely on land surface combustion models and Earth observations, atmospheric fire emissions can also benefit from remote sensing methods for detecting fire plumes and assessing their CO concentrations, as

demonstrated by the TROPOMI sensor (Zhou et al., 2022). This remote sensing data can be correlated with FRP (Griffin et al., 2023) and combustion efficiency (Van Der Velde et al., 2021). While it is important to validate this satellite data with actual atmospheric measurements, it offers valuable insights to study the impact of fire events (Yilmaz et al., 2023). Recent developments in this field (Vernooij et al., 2022) include the use of Unmanned Aerial Vehicles (UAVs), primarily applied to



grasslands and savannas. This approach is particularly promising for assessing the seasonal variability of emission factors
(Vernooij et al., 2021). However, this measurement technique is restricted over forests, especially in Europe, where safety
rules prevent the operation of aircraft or UAV's during firefighting interventions.

Our findings underscore that atmospheric tower measurements, while currently underutilized, represent an efficient and
consistent surrogate, particularly for CO emissions (Wiggins et al., 2021). We have demonstrated the critical role of MCEs
captured by the atmospheric mixing ratios in detecting smoldering combustion. Leveraging this information, we have enhanced
the existing generic fire emissions assessments for Europe under the Copernicus framework using the GFAS protocol (Kaiser
et al., 2012). This enables our bottom-up approach to be confronted and evaluated against atmospheric MCEs, an independent
approach to detect and identify fire behaviors.

The routine integration of these atmospheric data in future research holds the potential to unveil temporal patterns of flaming
vs. smoldering combustion within fire events and across different seasons, in line with recent observations collected across
various ecosystems (Carter et al., 2020; Zheng et al., 2018). Such an endeavor requires atmospheric inversion modeling due
to the distance from the actual combustion source, with plume dynamics influenced by wind direction, which could introduce
uncertainties related to meteorological data (Challa et al., 2008). Additionally, further investigations into emissions factors for
other greenhouse gases in the context of distinct fire types are warranted.

### 4.4 The 2022 fire-induce carbon emission budget

In our study, we took the year 2022 as a reference, a year marked by significant fire events in various ecosystems across
France, which are representative of Western Europe. A previous analysis conducted by Vallet et al. (2023) had already noted
a substantial increase in biomass loss during 2022 in France; primarily due to an expanded burned area across the country.
However, those conclusions were somewhat mitigated by the significant contribution of the low aboveground biomass affected
by fires in Mediterranean shrublands and young managed forests in Les Landes. It is worth noting that this previous study
provided an estimate solely for potential aboveground biomass loss.

In our research, we extended the analysis to account for soil combustion, which we identified through MCE measurements
from atmospheric towers. Consequently, our findings suggest that 7.95 ($\pm$ 3.63) MteqCO$_2$ were emitted into the atmosphere
during the 2022 fire season. Notably, 54.3 ($\pm$ 9.9) % of these emissions originated from the belowground biomass, with 35.4
($\pm$10.4) % from peat and SOM, and 18.95 ($\pm$ 0.65) % from lignite. These latter processes are often overlooked in fire emissions
assessment. In comparison, our estimates are 2-fold higher than the GFAS estimate of 4.18MteqCO$_2$ (CO and CO$_2$), which
excludes these processes.

Consequently, fire represents a huge source of greenhouse gases. Considering that the national carbon footprint amounted to
403,8 MteqCO$_2$ in 2022, fire represents 1.97 % ($\pm$ 0.89) of french emissions of greenhouse gases into the atmosphere (Citepa,
2023). Moreover, as forest is estimated to sequester 27 MteqCO2 per year in the country, fire disturbance would represent a
reduction of 30 % in this carbon sink for this particular year.





One remarkable aspect of 2022 fire season was the distinct impact on vegetation types (broadleaf vs. needle leaf), with varying rates of soil carbon accumulation. Temperate forests, characterized by a slower decomposition rate compared to the warmer Mediterranean climate, harbor more substantial litter and SOM density (Kurz-Besson et al., 2006). Additionally, our analysis revealed that the 2022 fires affected 510 ha of peatlands, as referenced in the Corine Land Cover dataset, contributing to 2.6 -

3.9% of the total carbon emitted.

While carbon stock associated with charcoal or lignite is often ignored, located beneath the SOM layer, we demonstrated here that this contributor is significantly impacted during this unusual fire season. This particular combustion impacted 2,265 ha over the lignite mines in Les Landes, a phenomenon reported by local authorities and substantiated by our low MCE measurements. These low MCE values, which are challenging to account for based on biomass or SOM combustion alone,

indicate the occurrence of lignite fires that could take place over an extended period. This phenomenon, reminiscent of the 'zombies' fires recently observed, has been reported by local authorities to have lasted even longer than expected over the winter 2022-2023 (McCarty et al., 2021; Irannezhad et al., 2020; Scholten et al., 2021; Kuklina et al., 2022). While lignite fires remain infrequent and typically omitted in carbon emissions inventories, they have been documented in other parts of the world (Stracher and Taylor, 2004; Brown, 2003; Fredriksson, 2004). These fires should raise concerns from authorities with

additional preventive measures in France, especially in areas with superficial lignite deposits and accumulated carbon residues from historical charcoal basins, some of which have grown to a substantial height of 100m in northern France (Anon, 2023).

Hotspot thermal anomalies and reignitions may persist up to three weeks after a fire, potentially emitting more carbon than our direct estimates suggest. These emissions, however, may be of a long-lasting nature but with a low intensity below the detection level of detection methods using atmospheric mixing ratios. Therefore, it is advisable to establish a more comprehensive

measurement network to better understand and to document this unexplored aspect of fire impact across European temperate forests.

Our results, while providing a preliminary and potentially conservative assessment of soil combustion in the region, underscore the need for enhanced field assessments of fire-induced effects on soil carbon stocks, particularly in peatlands and pine forests. These impacts could be even more substantial than initially calculated, emphasizing the importance of further investigation.

**4.5 Future directions for soil combustion modeling in Europe**

Our investigation into fire emissions during the 2022 fire season in France carries significant insights that can be extended to applications across the entire European continent. Current global fire emission assessments, such as GFED, GFAS, and FINN, predominantly focus on the combustion of deep SOM in boreal regions and specific tropical peatlands. In contrast, regions like European temperate forests and, by extension, our study area, are generally assumed to leave the soil unaffected by fire,

except for litter burning (Van Wees et al., 2022).

One limitation in existing greenhouse gas emission inventories from fires is the failure to adequately account for the transition between the flaming and smoldering phases in aboveground biomass combustion. Following a study on fire emissions in California, Mebust et al. (2011) cautioned that current emission factors might overestimate the contribution of flaming



combustion while underestimating the significance of smoldering combustion in total fire emissions. A concern also raised by

Garcia-Hurtado et al. (2013) in Europe, who estimated that 25% of emissions were associated with flaming and 75% with smoldering. Our approach sought to address this limitation by considering these different combustion phases in our processing chain.

A second limitation in current carbon emission inventories pertains to the SOM accumulation and combustibility, which may have been previously underestimated. Recent studies have identified significant instances of smoldering combustion in areas

where it was not previously considered, such as China's temperate forests (Tang et al., 2023) and even in African savannas towards the end of the burning season (Zheng et al., 2018). While temperate forests, characterized by milder temperatures and seasonal variations in soil moisture, were traditionally assumed to accumulate less carbon in soils compared to boreal forest, the actual situation is more nuanced. SOM levels (but also bulk density allowing for oxygen transfer and better combustion) can vary locally in Europe, depending on factors like local climate and specific soil and leaf types. These traits, such as pH

(Xiang et al., 2023) and leaf types (needles vs. broadleaves) can influence decomposition rates (Masuda et al., 2022; Krishna and Mohan, 2017; Cornelissen et al., 2011), highlighting the potential of using key plant traits as surrogates for SOM assessment. While SOM databases remain somewhat uncertain (Lin et al., 2022) insights from plant traits can be valuable.

The assumption that Mediterranean soils have been widely reported to hold low carbon stocks, thus not contributing to carbon emissions during fires, might not apply uniformly. For example, Certini et al. (2011) report that most carbon losses in

Mediterranean pine forests (Tuscany, Italy) are attributable to the elimination of the litter layer, rather than changes in the underlying mineral soil carbon content ; a conclusion also supported by Almendros and González-Vila (2012). This assumption might be actually true for broadleaf forests and shrublands, representing a large portion of burned area in Europe. However, smoldering combustion has been reported in some Mediterranean pine forests in Spain (Prat-Guitart et al., 2016), central European scots pines, and in California for upper and lower duff (Garlough and Keyes, 2011), with moisture thresholds of

57% and 102% (Hille and Den Ouden, 2005). Our study confirmed smoldering combustion in temperate Pine woodlands and heathlands. Therefore, we suggest that plant species distribution, and their leaf traits like pH and leaf type could be used to identify locations with substantial SOM accumulation, potentially leading to soil smoldering phases that should be included in carbon emission models. Notably, in higher latitudes (Turetsky et al., 2011b; Mekonnen et al., 2022; Walker et al., 2020) and eastern EU regions (Kirkland et al., 2023), carbon emissions from soil combustion can account for up to 90% of the total

carbon emitted. This has implications for the refinement of air quality estimates, which often rely on emissions derived from standard remote sensing information and models (Menut et al., 2023).

We recommend the initiation and compilation of an emission factor inventory over Europe, following initiatives in the US and Canada (Prichard et al., 2020). Additionally, considering duff peat emissions and making more extensive use of the atmospheirc tower network and fine temporal resolution remote sensing would enhance our understanding of fire events. Based

on the boreal and tropical experience, peatland moisture content appears to be a critical factor influencing combustion depth and emission factors. Smoldering of biomass at lower moisture contents develops wider pyrolysis fronts that release a larger fraction of other gas species (Rein et al., 2009). Pyrolysis can even reach very lower MCEs with large CO emissions (Song et



al., 2020; Kohlenberg et al., 2018) when temperatures reach above 400°C. Comprehensive models should integrate on-site peat and SOM moisture to account for changes in combustion rate and emission factors. This information has been available in France since 2016 through the peatland observation network (Bertrand et al., 2021; Gogo et al., 2021).

Understanding and predicting SOM and peat fire ignition and spread in temperate forests remain relatively unexplored areas of research due to the limited number of fire events as case studies. For instance, the ignition probability for SOM layers and peatlands is actually not yet fully comprehended. Pine cones have been identified as potentially influencing the ignition of soil duff (Kreye et al., 2013), thereby favoring smoldering, which is particularly relevant given that coniferous ecosystems tend to accumulate more SOM. Moreover, the spread of smoldering combustion is not well represented in current fire models, and its link with duff depth is minimal (Miyanishi and Johnson, 2002). The overall consequences of soil smoldering combustion extend beyond carbon emissions, affecting ecological factors, such as the regeneration potential of seeder species like pines (Madrigal et al., 2010, Watts and Kobziar, 2013). Consequently, we echo the conclusion reached by Xifré-Salvadó et al. (2020) that SOM and peatland fires in France and European temperate forests should be more deeply considered in terms of wildfire hazard, in particular for re-ignitions. For instance, the Landiras1 fire exhibited smoldering combustion for 10 days before reigniting from its south-western part over the lignite fires to ignite the Landiras 2 fire. Moreover, soil fires should be accounted for in forest planning and management, including soil fuel breaks strategies to halt smoldering combustion (Lin et al., 2021), in addition to the conventional focus on canopy fuel breaks.

## 5 Conclusion

This study offers compelling direct evidence of variable smoldering combustion rates observed during the atypical 2022 fire season. We employed the Modified Combustion Efficiency ratio, with atmospheric $CO_2$ and CO concentrations, calculated using data from the greenhouse gas atmospheric tower network situated throughout France. This particular year witnessed a significantly higher extent of burned area in the temperate Atlantic forest, marking a critical study case encompassing all major French sylvo-regions. Our findings allow us to draw several important conclusions :

First, we provided empirical support for the occurrence of soil and peatland fires, phenomena that have previously been insufficiently demonstrated or evaluated through remotely sensed burn area data

Second, we highlighted the large contribution of these fires within the overall carbon emission budget and trace gas emissions, which have not been fully integrated into existing fire emissions models.

Lastly, our study enabled us to propose valuable warning signals for assessing re-ignition hazards and developing post-fire management strategies based on the duration and intensity of hotspots within the affected area.

This research serves as a stepping stone for the development of future fire impact warning systems and emphasizes the potential of utilizing atmospheric greenhouse gas measurements in fire impact assessments. We also stress the imperative need for enhanced vegetation and soil carbon emissions factors during both flaming and smoldering phases. Finally, we advocate for




the widespread use of our updated fire emissions processing chain for France, which could potentially be extended to other

European temperate forests.

## Data availability

Fire model emissions are available through the OSU OREME website.

## Author contributions

LV, FM and TL supervised the study framework. LV performed data curation and analysis on the fire emission model. LV,

FM and PC assembled the fire emission model and parameters. CA, LJ and TL performed mixing ratios analysis. MR, ML and IXR provided data from the atmospheric towers. LV, FM and CA wrote the manuscript. All authors revised the manuscript.

## Competing interests

The contact author has declared that none of the authors has any competing interests.





**Figure 1. Map of the French forests with the location of fires larger than 30 ha that occurred in 2022 fire season. France is divided into four regions ('Atlantic Temperature forest', 'Atlantic pine forest', 'Mediterranean forest' and 'Other forest area') according to forest type (IFN, 2023b) and frequency of fire disturbance (BDIFF, 2023). The locations of the atmospheric towers (including ROC: Roc'h Trédudon, BIS: Biscarrosse, and OHP: Observatoire de Haute Provence) and the burned areas of the three corresponding main fires of interests are also represented ('Monts d'Arrée', 'Landiras 1' and 'Montagnette', red circles).**





**Figure 2. Refined fire emission model for temperate forest. The processing chain takes initial datasets as inputs to obtain exposure (burned area affecting each pool) and pool estimation (total amount of dry matter located in the burned area). Through specific values of Combustion completeness (CC), Smoldering fraction (SF) and Emission factors (EF), the model calculate combusted matter (fraction of pool actually combusted) and emissions to the atmosphere (CO and $CO_2$) in the flaming and smoldering phases (see Table 2).**



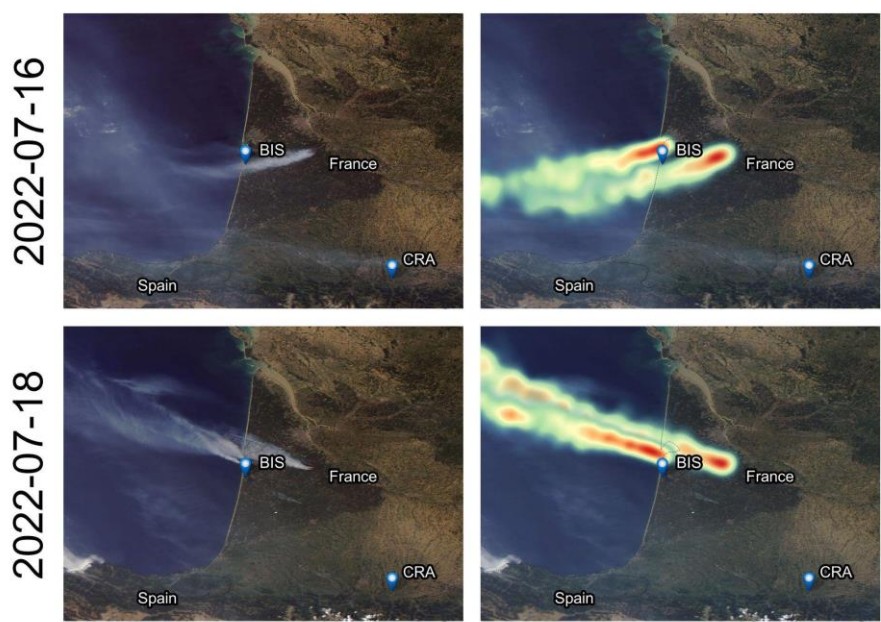

**Figure 3. Overlay of the MODIS (observed, left column) and the HYSPLIT (modeled, right column) plumes on16 and 18 July 2022 during the Landes wildfires (red for the highest particle density, yellow for the lowest particle density).**

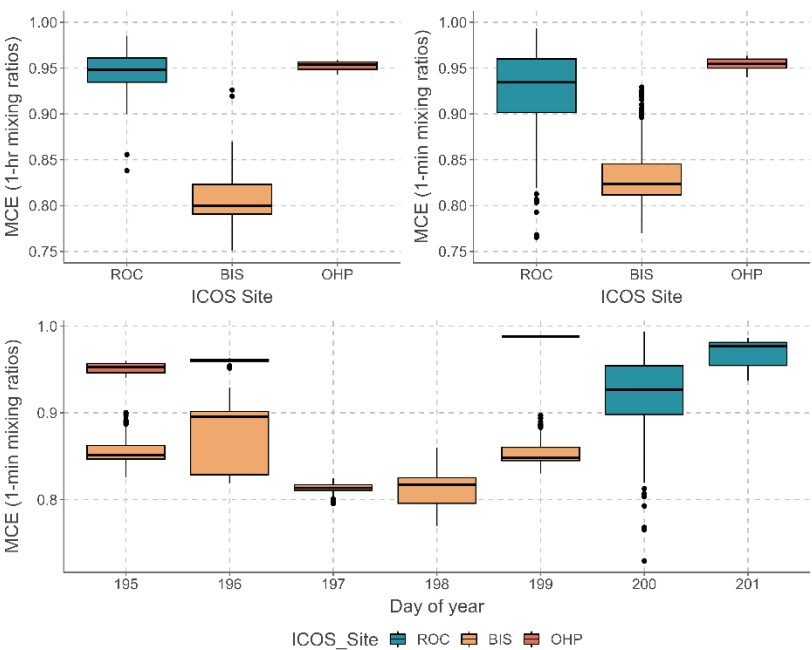

**Figure 4. Top : Median and quartiles of the Mean Combustion Efficiency (MCE) observed at the three atmospheric stations (ROC, BIS, OHP) impacted by the nearby fires Monts d'Arrée, Landiras1, and La Montagnette, respectively. The left graph shows 1-hour mixing ratios and the right graph shows 1-minute mixing ratios. Bottom : Daily median and quartiles values of the same corresponding data for 1-minute mixing ratios.**





**Figure 5. Top : Hotspot density (nb.ha⁻¹) for each main fire and its corresponding flux tower (BIS, OHP, ROC) and an example of hotspot distribution on BIS fire (Landiras 1), with corresponding Day of Year (DOY). Bottom : Median fire spread (km.h⁻¹) for each main fire and its corresponding flux tower (BIS, OHP, ROC) and an example of interpolated fire spread on BIS fire. The color scale indicates the day of the year of burning (decimal DOY) and arrows indicate the direction and rate of spread (proportional length of the arrow). Ignition corresponds to the pixel with the earliest DOY. We observed the change in spread direction toward south-west at first then moving west and north-west in accordance to changes in wind direction occurring during this fire (cf Fig. 3).**





**Figure 6. National footprint of France for the 2022 fire season. The Burned area (ha), Combusted matter (MtDM), CO2 and CO (Mt) emissions are shown for each region, each stock type (AGS : Aboveground stock, BGS : Belowground stock) and each pool. Values are provided in Table A1.**




**Table 1. Summary of the random forest model's performance across the atmospheric stations. The performance metrics are correlation coefficient (R) and root-mean-square error (RMSE). Tower location and height is also included.**

| Tower short name | Location | Height (AGL, m) | RF performance | | | |
| --- | --- | --- | --- | --- | --- | --- |
| | | | CO | | $CO_2$ | |
| | | | R | RMSE (ppb) | R | RMSE (ppm) |
| BIS | 44.38° N, -1.23° E | 73 | 0.87 | 9.04 | 0.98 | 1.12 |
| CRA | 43.13° N, 0.37° E | 60 | 0.87 | 9.05 | 0.98 | 1.35 |
| MDH | 49.24° N, 4.06° E | 48 | 0.86 | 8.43 | 0.99 | 1.56 |
| OPE | 48.56° N, 5.5° E | 50 | 0.88 | 7.18 | 0.98 | 1.22 |
| PUY | 45.77° N, 2.97° E | 10 | 0.88 | 6.61 | 0.99 | 0.82 |
| ROC | 48.41° N, -3.89° E | 80 | 0.92 | 5.85 | 0.99 | 0.65 |
| SAC | 48.72° N, 2.14° E | 100 | 0.89 | 8.62 | 0.98 | 1.34 |
| TRN | 47.96° N, 2.11° E | 50 | 0.89 | 6.41 | 0.98 | 1.16 |

**Table 2. Synthesis table of parameters used in the refined fire emission model. Minimum and maximum combustion completeness (CC), smoldering fraction (SF) and emission factors (EF) for the smoldering (S) and flaming (F) combustion to CO and $CO_2$ are based on previously reported values in the carbon emission scientific literature.**

| Stock and pools | CC min | CC max | SF | EF (g of gas per kg of DM pool) $CO_2$ F | $CO_2$ S | CO F | CO S | references |
| --- | --- | --- | --- | --- | --- | --- | --- | --- |
| **Aboveground stock (AGS)** | | | | | | | | |
| stem | 0.10 | 0.50 | 0.40 | 1,700 | 1,400 | 73 | 165 | (Van Wees et al., 2022; Prichard et al., 2020; Balde et al., 2023; Akagi et al., 2011) |
| branch | 0.90 | 1.00 | 0.00 | 1,686 | | 63 | | (Van Wees et al., 2022; Prichard et al., 2020) |
| leaf | 0.90 | 1.00 | 0.00 | 1,686 | | 63 | | (Van Wees et al., 2022; Prichard et al., 2020) |
| shrub | 0.40 | 0.99 | 0.40 | 1,746 | 1,460 | 72 | 93 | (Van Wees et al., 2022; Prichard et al., 2020; Akagi et al., 2011; Garcia-Hurtado et al., 2013) |
| grass | 0.90 | 1.00 | 0.00 | 1,686 | | 63 | | (Van Wees et al., 2022; Prichard et al., 2020) |
| litter | 0.80 | 1.00 | 0.10 | 1,696 | 1,750 | 64 | 119 | (Van Wees et al., 2022; Prichard et al., 2020) |
| **Belowground stock (BGS)** | | | | | | | | |
| SOM | 0.10 | 0.50 | 0.90 | 1,696 | 1,000 | 64 | 298 | (Van Wees et al., 2022; Prichard et al., 2020) |
| peat | 0.05 | 0.20 | 0.90 | 1,696 | 1,000 | 64 | 298 | (Van Wees et al., 2022; Prichard et al., 2020; Akagi et al., 2011; Rein et al., 2009; Geron and Hays, 2013) |
| lignite | 0,01 | 0,025 | 1.00 | | 1,500 | | 750 | (Song et al., 2020) |



**Table 3. Description of ROC, BIS and OHP fires in terms of exposure (ha of vegetation and soil types affected), pool dry matter density (tDM.ha⁻¹) for aboveground (stem, branch, leaf, shrub, grass, litter) and belowground (SOM, peat, lignite) pools, and the resulting total pool dry mass actually affected by fire (tDM).**

|  | ROC | BIS | OHP |
|---|---|---|---|
| **EXPOSURE (ha)** | | | |
| fire | 1,726 | 12,140 | 1,477 |
| forest | 129 | 8,622 | 1,124 |
| shrubland | 54 | 1,257 | 226 |
| grassland | 1,093 | 2,200 | 127 |
| soil | 1,276 | 12,078 | 1,477 |
| peatland | 449 | 61 | |
| lignite | | 1,909 | |
| **POOL DENSITY (tDM.ha⁻¹)** | | | |
| stem | 25.0 | 40.7 | 42.3 |
| branch | 8.5 | 13.8 | 14.4 |
| leaf | 12.9 | 5.7 | 3.7 |
| shrub | 7.8 | 7.3 | 10.0 |
| grass | 4 | 4 | 4 |
| litter | 5.0 | 7.3 | 3.8 |
| SOM | 140.1 | 235.7 | 95.2 |
| peat | 2,900.0 | 2,900.0 | |
| lignite | | 14,000 | |
| **POOL DRY MASS (tDM)** | | | |
| stem | 3.22e+03 | 3.51e+05 | 4.75e+04 |
| branch | 1.10e+03 | 1.19e+05 | 1.62e+04 |
| leaf | 2.36e+03 | 5.61e+04 | 4.97e+03 |
| shrub | 4.23e+02 | 9.16e+03 | 2.26e+03 |
| grass | 4.43e+03 | 8.84e+03 | 5.21e+02 |
| litter | 6.34e+03 | 8.79e+04 | 5.64e+03 |
| SOM | 1.79e+05 | 2.85e+06 | 1.41e+05 |
| peat | 1.30e+06 | 1.77e+05 | |
| lignite | | 2.67e+07 | |






**Table 4. Bottom-up approach from stock to carbon emissions. Total pool dry matter combusted (tDM) and CO₂ and CO emissions (in g) estimates are based on parameters of Table 2. The resulting MCE is provided for each approach (considering only AGS or including also BGS), each fire and each combustion phase. AGS : Aboveground stock, BGS : Belowground stock, FP : Flaming phase, MP : mixed phase, SP : Smoldering phase.**

|  | Stock type | Matter combusted (tDM) | Emission (g) | | MCE |
|---|---|---|---|---|---|
|  |  |  | $CO_2$ | CO |  |
| **AGS ONLY** |  |  |  |  |  |
| ROC | AGS | 1.45e+04 (± 1.8e+03) | 2.44e+10 (± 2.97e+09) | 9.99e+08 (± 1.5e+08) | 0.961 (± 0.001) |
| BIS | AGS | 3.66e+05 (± 9.09e+04) | 6.06e+11 (± 1.46e+11) | 2.86e+10 (± 9.11e+09) | 0.956 (± 0.004) |
| OHP | AGS | 4.15e+04 (± 1.18e+04) | 6.84e+10 (± 1.89e+10) | 3.34e+09 (± 1.2e+09) | 0.955 (± 0.004) |
| **AGS + BGS** |  |  |  |  |  |
| ROC |  |  |  |  |  |
| FP |  |  |  |  | 0.961 (± 0.001) |
|  | AGS | 7.23e+03 (± 8.99e+02) | 1.22e+10 (± 1.48e+09) | 4.99e+08 (± 7.49e+07) |  |
| MP |  |  |  |  | 0.828 (± 0.015) |
|  | AGS | 7.23e+03 (± 8.99e+02) | 1.22e+10 (± 1.48e+09) | 4.99e+08 (± 7.49e+07) |  |
|  | BGS | 5.41e+04 (± 3.34e+04) | 5.79e+10 (± 3.57e+10) | 1.49e+10 (± 9.16e+09) |  |
| SP |  |  |  |  | 0.796 (± 0.001) |
|  | BGS | 1.62e+05 (± 1e+05) | 1.74e+11 (± 1.07e+11) | 4.46e+10 (± 2.75e+10) |  |
| BIS |  |  |  |  |  |
| FP |  |  |  |  | 0.956 (± 0.004) |
|  | AGS | 1.83e+05 (± 4.54e+04) | 3.03e+11 (± 7.29e+10) | 1.43e+10 (± 4.56e+09) |  |
| MP |  |  |  |  | 0.821 (± 0.015) |
|  | AGS | 1.83e+05 (± 4.54e+04) | 3.03e+11 (± 7.29e+10) | 1.43e+10 (± 4.56e+09) |  |
|  | BGS | 3.36e+05 (± 1.96e+05) | 4.1e+11 (± 2.31e+11) | 1.48e+11 (± 7.76e+10) |  |
| SP |  |  |  |  | 0.729 (± 0.011) |
|  | BGS | 1.01e+06 (± 5.87e+05) | 1.23e+12 (± 6.93e+11) | 4.44e+11 (± 2.33e+11) |  |





**Appendix**

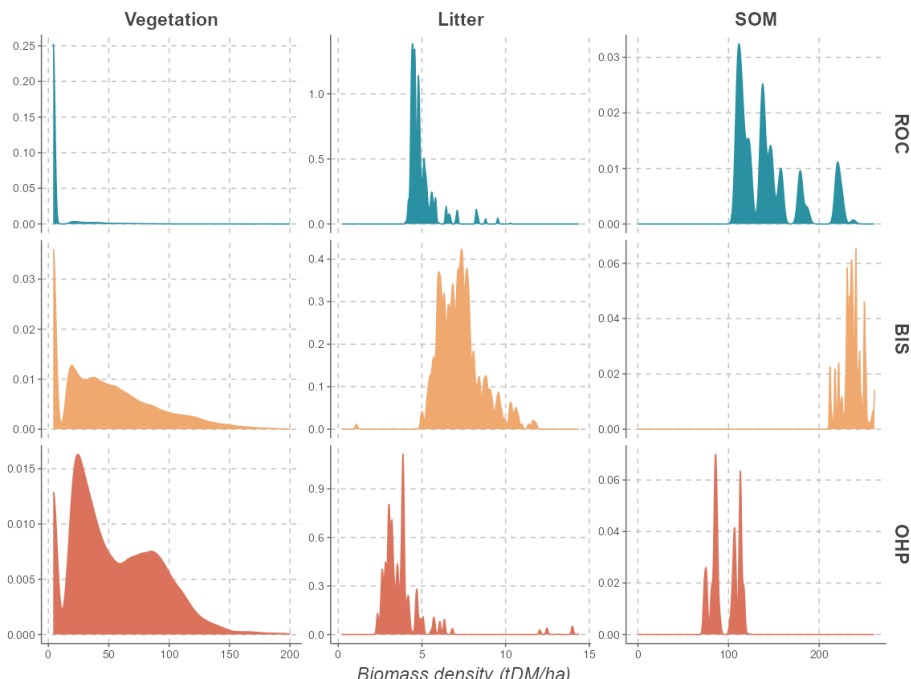

**Figure A1. Vegetation biomass (stem, branch, leaf, shrub and grass), litter and SOM density (tDM.ha-1) distribution for the BIS, ROC and OHP fires.**

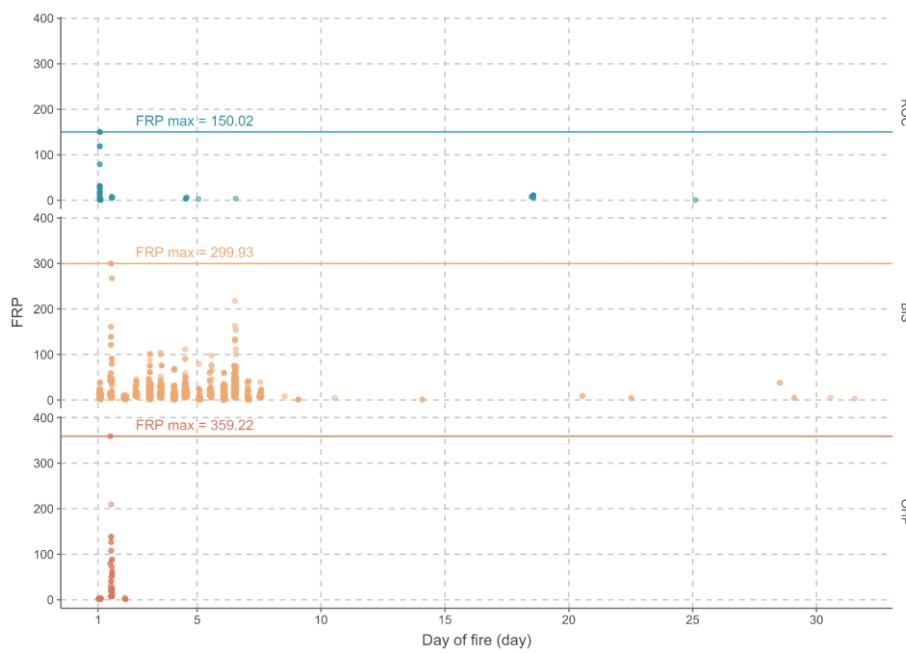


**Figure A2. VIIRS/MCD14ML Fire Radiative Power (FRP, in MW) temporal distribution from ignition to 5 weeks after ignition for each ROC, BIS and OHP fires.**



**Table A1. Burned area (ha), Stock (MtDM), Matter combusted (MtDM), CO$_2$ and CO emissions (in Mt), resulting mean MCE, and GFAS estimation in France for the 2022 summer fire season and for the 4 regions.**

| *Region* | *Burned area (ha)* | *Stock type* | *Stock (MtDM)* | *Matter combusted (MtDM)* | *Emission (Mt)* | | *MCE* | *GFAS Emission (Mt)* | |
|---|---|---|---|---|---|---|---|---|---|
| | | | | | CO$_2$ | CO | | CO$_2$ | CO |
| Atlantic Temperate forest | 2,315 | AGS | 0.081 | 0.052 (± 0.010) | 0.086 (± 0.017) | 0.004 (± 0.001) | 0.841 (± 0.017) | 0.155 | 0.007 |
| | | BGS | 1.546 | 0.236 (± 0.146) | 0.252 (± 0.156) | 0.065 (± 0.040) | | | |
| Atlantic Pine forest | 26,850 | AGS | 2.351 | 1.278 (± 0.350) | 2.111 (± 0.559) | 0.102 (± 0.036) | 0.834 (± 0.015) | 2.914 | 0.159 |
| | | BGS | 38.121 | 2.447 (± 1.498) | 2.856 (± 1.704) | 0.936 (± 0.524) | | | |
| Mediterranean forest | 7,600 | AGS | 0.332 | 0.199 (± 0.046) | 0.330 (± 0.074) | 0.015 (± 0.005) | 0.957 (± 0.003) | 0.272 | 0.014 |
| | | BGS | 0.850 | | | | | | |
| Other forest area | 4,839 | AGS | 0.590 | 0.315 (± 0.087) | 0.519 (± 0.139) | 0.025 (± 0.009) | 0.955 (± 0.004) | 0.516 | 0.024 |
| | | BGS | 0.808 | | | | | | |
| Total | 41,600 | | 44.680 | 4.526 (± 2.138) | 6.154 (± 2.650) | 1.147 (± 0.615) | 7.172 (± 0.081) | 3.857 | 0.204 |

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
