# Peer review of "Soil smoldering in temperate forests: A neglected contributor to fire"

_EGUsphere, 2023_

## Author Comment (AC1)

This paper combined satellite observations of fire behaviour, tower-based CO and CO2 mixing ratios and bottom-up approach to estimate forest fire emissions in France with a focus on improving emissions from smoldering. I found the methods used by the authors in general credible and the paper advances the quantification of fire emissions induced by forest fires. I have a few major comments mainly regarding clarifications of the methods being used and some minor technical comments (detailed below).

Major comments:

- I suggest adding a paragraph giving an overview of the methods, preferably with a flowchart figure, focusing on how different approaches are combined and connected.

  After a thorough reading of the paper, we recognize that this part needs further explanation. We have therefore produced a new figure in the appendix, and greatly modified section 3.5.

- What's the major purpose of Hysplit model? I don't really see how it is connected with the selection of tower sites and determination of background measurement…. Is it only used to justify that most of the ICOS sites are free of influences of Mediterranean forest fires and hence their measurement could be considered as background ones? Fig. 3 is nice but also quite unique I guess. Is it a sufficient example to argue that, based on Hysplit simulations, most of the ICOS sites are free of influences of Mediterranean forest fires and hence their measurement could be considered as background ones?

  **In this study, the Hysplit forward-trajectories were used to identify the arrival times of the forest fire plumes to the observation site. The results were cross-checked by checking for CO2/CO anomalies in the observation data happening at the same time. As for the back-trajectories, we modeled them to define the influence matrix describing the Source-Receptor relation on a 0.05 x 0.05 deg grid. The influence matrices were used to check for additional sources influencing each tower. We also used those two approaches to check the transport towards inland stations where spikes in CO2/CO were observed. The results showed that the plumes from both fires reached those stations at the same time.**

  **At first, we wanted to look into the data collected by the whole ICOS-France network. However, after checking the model results, we discarded the distant stations since the measured signals turned out that most detected plumes corresponded to multiple overlapping plumes (i.e. multiple fire plumes reaching the ICOS station at the same time). Also, as stated previously, we used the model data to ensure that no other sources could be linked to the signals we detected (e.g. cities, power plants, ecosystem respiration). This was the case of the OHP station where many signals were discarded because of their potential anthropogenic origin.**

  **We added this explanation to the text as follows:**

  > **This step was accomplished using the Hybrid Single Particle Lagrangian Integrated Trajectory (Hysplit) model (Stein et al., 2015). In a backward-in-time configuration, particles were released from the receptor site and monitored over 7-day intervals. The result is a footprint matrix representing the influence of the area around the receptor on the measurements. The model spatial resolution used is 0.05 x 0.05 deg. The Global Forecast System (GFS) meteorological model (National Centers For Environmental Prediction/National Weather**

**Service/NOAA/U.S. Department Of Commerce, 2015) provided the atmospheric conditions (wind and turbulence) to drive these particles from the receptors to the sources in the Hysplit simulations. The GFS outputs, featuring a horizontal resolution of 0.25° x 0.25° and 3-hourly time intervals, served as the meteorological inputs. We also conducted Hysplit simulations in a forward-in-time configuration releasing particles (600 per hour) from the fire locations, over the fire duration from the exact burned area. In this configuration, we simulated the transport of the plume from the fires to the ICOS stations. By tracking the arrival times of the fire-emitted particles within an influence region surrounding each atmospheric tower, we successfully attributed a fire source to each anomaly.**

- If I understand well, it seems that the characterization of fire behaviour using satellite data is independent of the bottom-up estimation of fire emissions, in particular, the fire behaviour information has not been used to determine the key parameters in Equation (2) (e.g., SFp) and parameters in lines 330-334.

  **Our first hypothesis was indeed that rate of spread could be an index discriminating soil smoldering fires. However, as stated Line L430-431 this was not the case, as the ROC fire in the temperate forest and affecting the peatland was fast-spreading according to hotspots. To consider which fire would experience smoldering combustion other than the 3 fires from which could measure MCE, we relied on fire duration based on the detection of hotspots within the fire polygon after the initial spreading leading to the final fire shape up to 5 weeks after ignition. Figure A2 illustrates that the discriminating factor between the ROC and BIS fires experiencing smoldering combustion and the OHP mediterranean fire is the long lasting detection of hotspots. We selected this index to consider SOM combustion or not in our study. L645-666 we discussed what could be a discriminating factor for considering SOM smoldering in temperature Europe. Beside peatlands references in land cover datasets, we ended up with suggesting the slow decomposing needleleaf to be a major discriminating factor under mild and wet climate conditions limiting their decomposition and accumulating SOM. Unfortunately, fully detecting smoldering fires remain impossible yet from remote sensing, and our result more warn about potential underestimation of fire emission in temperate ecosystem than propose a defined criteria to identify actually lead to soil smoldering or not.**

- This is somewhat a little disappointing. Following this logic, it then seems that the key strength/advancement of the paper is that the authors compiled a nice Table 2, a range of more credible sources of fuel load, and the used satellite-derived fire information and power-based MCE to *indirectly* verify their bottom-up estimate of emissions? Is this correct? This point has to be made clearer when the authors address my first major comment.

  **Indeed, fire spreading was not a convincing indicator. the presence of peatlands, hotspot duration after the fire spread leading to the final fire shape and leaf type (needles) are our main hypothesis that we used to calculate SOM smoldering or not. we better explain this step in the flow chart.**

- The authors examined three typical fires, or fires in three typical forests using satellite-based fire behaviour and power-based mixing ratio measurements. These are then used to support

their bottom-up approach. Then then the challenge is how we can ensure that the upscaling to the national level using their bottom-up approach is also reliable, given that fires are highly temporal and spatially heterogeneous in terms of fire bebaviour, fraction of flaming versus smoldering, combustion completeness etc. (I believe the authors have tried to address well the spatial heterogeneity in fuel load)?

**We provided here a conservative approach to implement SOM and peatland smoldering combustion compared to GFAS fully omitting this component. We could identify from the 3 fires tested that hotspot duration, the presence of peats and the leaf type (needleleaf) associated to high SOM content (calculated from GFED) would be the information to upscale our finding to the other fires. other smoldering fires might happen under other conditions that we could not identify yet. our work contributes to a first attempt, based on flux tower evidences, in starting and implementing smoldering fires in temperate forests, neglected until now. We cannot ensure a full reliability that other smoldering fires did not happen, but we believe our selection criteria were conservative not to overestimate SOM smoldering. No other fires than ROC and atlantic forest fires actually met the criteria in our study.**

Minor comments:

Line 139: some introduction on VIIRS data is necessary because it seems an important limitation on what fires have been analyzed.

**We better explain what is VIIRS, its spatio-temporal resolution and performance in detecting fires now in paragraph 2.2.1 (L140-145).**

Line 145: "beyond the fire outbreak ". What does 'beyond' mean here?

**we rephrased the sentence by using 'after the fire ignition date ' instead of 'beyond fire the fire outbreak'.**

Line 146–147: I don't see how the approach described here (visual examination of RGB spectrum) could be reconciled with BAMTS… So what is exactly the role of BAMTS in burned area detection? And how are these two further linked with random forest classifier and how the classifier is used and for which purpose?

**we fully rephrased L153-160 paragraph 2.2.1 describing this keystone step. as a first step, BAMS automatically generates RGB spectral differences between the pre and post fire period over the study region and according to the ignition date fixed by the user. the user sets up (by visual examination of the spectral difference map) manually the burned and unburned training zone as two samples of the study region. the random forest classifier (as part of the BAMTS tool) then reclassifies the whole study as burned or unburned. omissions and commissions errors can happen in this first step, so the user can enlarge the training regions (to cover the commission error and include them in the 'unburned' training zone for example) to reach the cleanest fire polygone as possible. This final step is also dependant on a visual inspection of commission and omissions errors based the spectral difference map.**

Line 203: "corresponding to a single grid cell. ". Which model does this grid cell refer to? What is the spatial resolution of Hysplit?

**This statement refers to the hysplit grid cell which has a 0.05 x 0.05 deg resolution. The statement was modified and the following sentence was added to the text.**

*The model spatial resolution used is 0.05 x 0.05 deg.*

Table 1: Better to report R2 rather than R. The same for the texts.

**As per the suggestion of the reviewer, we replaced R by R2**

Line 168: what is this 6-hour data?

Hotspot data (thermal anomalies) are obtained with VIIRS sensors (S-NPP and NOAA) and MODIS sensors (Aqua and Terra) **The time overpass varies between 3 and 9hours that we averaged at 6hours. We now provide the true range.**

Line 199: Is this 600 per hour particle numbers typically used in transportation modeling? How does this influence the results?

**The is no typical standard number of particles to release. We gradually increased the number of released particles until the full extent of the fire plumes was reached. Larger release rates were not producing any additional information. We didn't include this part of the work to avoid a long technical description.**

Line 201: "By tracking the arrival times of these particles within an influence region surrounding each atmospherictower, we successfully attributed a source to each anomaly", I don't understand the latter half. Could you please explain?

**The role of the Lagrangian model in this study is to determine the source-receptor relationship between the forest fires and the CO and CO2 peaks in the observed data. In the forward in time mode, we released the particles from the fires locations and tracked them in space and time to determine if the high CO and CO2 values observed at a certain ICOS tower correspond to the arrival of the plume from a fire. In the model, this corresponds to the particles arriving at a predefined influence region around the ICOS tower. This is important to identify other potential sources of anthropogenic origin that could influence the results of this study.**

Line 298-299: I don't understand what you mean by 'baseline' here.

**we rephrased the sentence L298**

Table 2: I cannot reconcile/connect Table 2 with lines 330–335. (1) you provide only constant SF values in Table 2. But if SF values do no change among the flaming phase, mixed phase and smoldering phase, then how is this used in Equation (2)? (2) lines 330-335 seems giving proportions of fuels being affected by fire, what is the difference between this and CC in Table 2? Seems that lines 330-335 should be better integrated with Table 2 so that you have only a single source to present the parameters used in emissions calculation. (3) how the information in lines 330-335 is used in Equation (2)? (4) how do you choose CC values between its min and max values in Table 2?

**we names the three phases flaming, mixed and smoldering, which brought confusion to the interpretation of the results. we now name these phases respectively as spreading,**

mixed and post-spreading stages, as both flaming and smoldering can happen during the 3 stages. during the spreading stage (early after ignition) we considered that 50% of the burned area is affected by flaming, while soil smoldering did not start yet.  during the mixed stage, we considered that the remaining 50% of the burned areas is still under flaming while the previous 50% enter the smoldering stage and affecting 50% of the soil c stock (so 25% of the total C stock combustion during the whole fire duration). during the post-spreading stage, we considered no more flaming, and smoldering combustion consuming the remaining 75% of the total soil C stock affected during the whole fire duration. in turn, summing up the 3 stages, leads to the usual emission estimates of total C stocks affected by combustion but calculated all at once. This splitting along the fire progression could capture the temporal dynamic of MCEs and capture the very low MCEs that we could only attribute to lignite high CO emission factors according to the EF synthesis from table 2.

**We rephrased the description of this part in the manuscript with using this renaming of stages to prevent confusions.**

Line 352: TROPOMI data not explained in Methods.

**Since the data was not used in this study, we did not provide a description of it.**

Figure 4: what is the difference between 1-hour and 1-minute? Are they the temporal resolutions of the data ? what is the temporal resolution of measurement over the towers?

**The Picarro data from the ICOS sites are available at 1-hour and 1-minute temporal resolutions. When multiple sampling heights are present on a single ICOS tower, the 1-hour interval is split equally between all levels. Thus, for the towers with three levels the 1-hour data is an aggregation of almost 15-20 minutes only. This along with the ability of the 1-minute resolution to capture faster variations of the mixing ratios, motivated the comparison between both resolutions.**

---

## Author Comment (AC2)

Firstly, many apologies for the delay in providing my review. But, this proved a very interesting manuscript which I enjoyed reading in detail. The authors present analysis of atmospheric CO2 and CO measurements from ground stations influenced by smoke plumes from large fires in France in 2022, demonstrating the presence of significant amounts of smouldering combustion in at least one of the three large fire events that took place that summer. They subsequently construct a sophisticated bottom-up fire emissions inventory for France which incorporates detailed land cover and carbon content data, and explicitly estimates the emissions contributions from both smouldering and flaming combustion. Explicit consideration of smouldering combustion appears to result in a better match with the observed combustion efficiencies than the GFAS emissions estimate for CO2 and CO from the same fires. Furthermore they find that due to the inclusion of data on belowground organic soil carbon stocks, the bottom-up emissions estimate predicts a ~2x larger total carbon emissions across France in 2022 than GFAS does, with implications for the country's carbon budget.

The manuscript highlights the very significant contribution that belowground carbon stocks can make to fire emissions in regions where smouldering combustion has not previously been considered important to include in global emission inventories, as well as how station measurements of atmospheric composition can be used to discern the relative importance of smouldering combustion from individual fires. It's highly relevant and should be of significant interest to the readership of Biogeosciences.

I have a few comments and suggestions which I have detailed below. It's a slightly long list because I was keen to really read and understand this manuscript in detail! But hopefully are mostly just clarification. There are some areas where I feel the methods require more explanation and there is some extra information needed to fully understand the process. Additionally, from a story-telling perspective the two strands of the manuscript (observations of Modified Combustion Efficiency using the atmospheric tower network, and bottom-up fire emissions estimation) could be linked together more strongly, and it would be nice to see them being compared and used to inform and validate the other more directly. Finally, I have some concerns that the uncertainties on some of the parameters used in the emissions estimation are being underappreciated, which may mean that the results need to be interpreted with a little more caution until the model can be explicitly validated with other observational data sources. I regard these all as minor revisions though, since they do not undermine the main objective or results of the manuscript, but perhaps just imply some additional thought on the presentation. (I've also listed some technical comments, which are mostly just small wording/grammatical suggestions). Subject to the authors addressing these points (or pointing out why I've misunderstood! Which is certainly possible) then I'd very much support the manuscript being accepted for publication.

**Minor comments:**

**L23:** "We examined the role of soil smoldering combustion responsible for higher carbon emissions, locally reported by firefighters but not accounted for in global fire emission budgets" – I think this statement is only partially true. Widely-used fire emission inventories like GFED and GFAS do include a representation of peat and/or organic soil as a distinct land cover/biome type (with spatial distributions of peat and organic soils based on the literature, much like in the present study). For instance, GFAS has an 'extratropical forest with organic soil' land cover class, and a 'peat' land cover class. Consequently, even if they do not necessarily have separate a smouldering versus flaming representation, they can still have different fuel consumption and/or emission factors calibrated for these land cover types, which reflect different carbon pools and contribution of smouldering combustion. They may well still underestimate the contribution of smouldering combustion, due to it being undetected or due to incorrect assumptions of the land cover/soil types over Europe, but that's not the same as it being entirely unaccounted for. N.B. the authors' discussion of this issue in Sect. 4.5 is very good, and much more nuanced.

after carefully reading our manuscript, we understand the confusion as we did not properly clarify our statement. We target here temperate and mediterranean forests covering France. In the description of GFED5 (van wees et al. 2022), the authors state that "Fire emissions from the combustion of SOC were only modelled for the boreal region and specific tropical peatlands, whereas in other regions the soil was assumed not to be affected by fire". We used this sentence to state that temperate forest were not cover for the combustion of SOC. the way we wrote our sentence might be confusing for the reader as we understand that SOC combustion is never calculated. We rephrased the sentence so that the readers more clearly understand that we talk here about temperate forests, and not about all biomes. thanks for raising this needed clarification.

**L105-106:** "We also compared our emissions to the current global models based on standard fire emission factors (GFAS, 2023)" – if you compared only with GFAS, then technically you have only compared against one current global model, not models. Also 'based on standard emission factors' is an interesting point to consider – does the authors' work here indicate that the GFAS biome-average fuel consumption and/or emission factors are wrong, or that the assumptions of what vegetation/material is burning is wrong? E.g. the 'standard emission factors' could be perfect, but if applied to an incorrect map of vegetation density or soil type, you could still infer the wrong total emissions.

**We rephrased to "We also compared our emissions to GFAS, (2023) emissions used by the Copernicus Atmosphere Monitoring Service (CAMS | Copernicus, 2023) as a reference dataset and publicly delivered in near real-time to stakeholders and society".**

**We considered here that emission factors are accurate and we have nothing in hand to counterbalance their emission factors, so we mostly focused our study on accounting additional combustion happening in soils.**

**L152-153:** "Among these fire polygons, three of them located in the proximity of atmospheric towers were chosen for in-depth analysis, referred to as "main fires"" – it's rather unclear, is all the subsequent analysis of the atmospheric tower data restricted to the time periods around these three 'main fires', or is the analysis carried out for all fire events but with some particular features discussed in more detail for these three fires? If the analysis is focused on these three fires then they need to be described in a little more detail: e.g. what dates did they start/finish, what area did they burn?

**Yes, we examined the MCE only for these three fires. Table 3 partially described the 3 fires in terms of burned area and vegetation affected. We now inserted their date of ignition and duration in Table 3. we also rephrased L152-153 to clarify that we studies emissions from towers only on these 3 fires.**

**Duration is however tricky, as the reported duration is the flaming, while firefighters reported continuous soil burning for months (ROC) and even years (still burning in 2024) for the BIS fires over lignite.**

**L168:** "we leveraged the temporally dated (6-hour intervals) spatial locations of fire hotspots (Fig. 5)" – as I understand it, the fire hotspots come from MODIS and VIIRS. However, Terra and Aqua overpasses are only 3 hours apart (10:30 am/pm and 1:30 am/pm respectively). S-NPP and NOAA-20 overpasses are also both around the same time as Aqua (1:30 am/pm). How were the fire hotspot detections converted to regular 6-hourly intervals? (Also, what happened to figures 2, 3, and 4? Figures should be numbered following the order they appear in the text; this is the 2nd figure referenced in the paper).

**true, so the time overpass varies between 3 and 9hours that we averaged at 6hours. We now provide the true range.**

**To ensure a logical flow in the appearance of the figures, we have removed the reference to figure 5 at this point.**

**L182-186:** "Data collection for this study spanned from June 15th to September 1st, 2022. In the context of the Atlantic pine forest, the dominant winds were from the northeast, propelling the plume seaward. Notably, a shift in wind direction occurred on July 14th -15th, with the wind veering to the north-northwest. This shift contributed to the highest CO peaks observed at the Biscarrosse station" – again, it's very unclear whether data is being analysed for multiple fires across the season (June – September), for just for the single large fire that occurred in this region ('the plume'). Was a fire event even taking place on July 14th -15[th] – the relevance of this date hasn't been explained.

**we now dated our 3 three main fires used for analysing emissions from flux towers in table 3. they started between july 12th to july 18th.**

**The fires in the Landes Forest started on July 12[th], northeasterly winds were dominating thus transporting the plume towards the ocean. On the 14[th], we observed a shift in the wind and the first signals from the fire were detected at the Biscarrosse station. Given the variability of the winds, the signals from the fires were not always detected by the ground stations, such was the case of the fires near OHP. We added the specifics to the corresponding sentence.**

*In the context of the Atlantic pine forest that started on July 12[th], the dominant winds were from the northeast, propelling the plume seaward. Notably, a shift in wind direction occurred on July 14[th]-15[th], with the wind veering to the north-northwest. This shift contributed to the highest CO peaks observed at the Biscarrosse (BIS) station.*

**L191-205:** This is probably just me not understanding very well, but: this paragraph describes both back-trajectory and forward-trajectory analysis to determine the sources corresponding to CO mixing ratio anomalies observed at each atmospheric tower. I'm not clear how these two approaches were combined to ultimately attribute a single source for each anomaly. What if there were multiple potential sources within the back-trajectory footprint of a tower? Also, did the authors only consider forest fire hotspots as potential sources? E.g. What if the wind direction on a particular day was blowing from an urban source which could also result in anomalously high CO - how were these possible contributions ruled out? Finally, how was the influence of a particular fire temporally co-located with the tower measurements? I.e. how did the authors decide over what time period did the measured MCE correspond to a particular fire?

**In this study, the Hysplit forward-trajectories were used to identify the arrival times of the forest fire plumes to the observation site. The results were cross-checked by checking for CO2/CO anomalies in the observation data happening at the same time. As for the back-trajectories, we modeled them to define the influence matrix describing the Source-Receptor relation on a 0.05 x 0.05 deg grid. The influence matrices were used to check for additional sources influencing each tower. We also used those two approaches to check the transport towards inland stations where spikes in CO2/CO were observed. The results showed that the plumes from both fires reached those stations at the same time. Furthermore, in the case of the OHP station, many signals were discarded because of their potential anthropogenic origin. We added this explanation to the text as follows:**

**This step was accomplished using the Hybrid Single Particle Lagrangian Integrated Trajectory (Hysplit) model (Stein et al., 2015). In a backward-in-time configuration, particles were released from the receptor site and monitored over 7-day intervals. The result is a footprint matrix representing the influence of the area around the receptor on the measurements. The model**

spatial resolution used is 0.05 x 0.05 deg. The Global Forecast System (GFS) meteorological model (National Centers For Environmental Prediction/National Weather Service/NOAA/U.S. Department Of Commerce, 2015) provided the atmospheric conditions (wind and turbulence) to drive these particles from the receptors to the sources in the Hysplit simulations. The GFS outputs, featuring a horizontal resolution of 0.25° x 0.25° and 3-hourly time intervals, served as the meteorological inputs. We also conducted Hysplit simulations in a forward-in-time configuration releasing particles (600 per hour) from the fire locations, over the fire duration from the exact burned area. In this configuration, we simulated the transport of the plume from the fires to the ICOS stations. By tracking the arrival times of the fire-emitted particles within an influence region surrounding each atmospheric tower, we successfully attributed a fire source to each anomaly.

**L321:** "Table 2 provides a comprehensive summary of CC, EF, and SF for each pool" – Table 2 lists both a min and max value of CC from the literature for each pool – but which value was actually used in the emissions calculation? This doesn't seem to be described. For some of the pools (particularly the belowground stocks), the choice of CC within the range reported will make a huge difference to the total C emissions, which doesn't seem to be acknowledged or included in the uncertainties given for the total emissions in Table 4 and Sect. 3.5.

In fact, the minimum and maximum CC values for certain pools can vary widely, resulting in associated uncertainty. The values presented in the tables (M, E, MCE) and in the text correspond to the average between the estimates with CCmin and CCmax. The range of uncertainty corresponds to the difference between this mean value and that obtained with limit CC. Maximum and minimum estimates have the same deviation from the mean, as only two values have been retained (min and max). For example, when we say that 7.95 MteqCO2 ($\pm$ 3.62) were emitted by fires in 2022, 7.95 corresponds to the average obtained between the estimates with CCmin (7.95-3.62=5.33) and CCmax (7.95+3.62=11.57). We have added this paragraph in order to clarify the calculation method, which was previously absent:

We provide a range of values for combustion completeness (CCmin and CCmax). The estimated values for combustion matter (M), emission (E) and MCE correspond to the average between the minimum and maximum estimates. The uncertainty ranges correspond to the deviation between this mean value and the limit value (min or max estimations having the same deviation from the mean).
 **We performed our calculations using the mean, min and max values of parameters from table 2 and got min, max and mean values for emissions. We actually mention this uncertainty in the section 3.5 and table A1 and figure 6. We rephrased some part of section 3.5 to emphasize more on this uncertainty.**

**L329-335:** "we delineated three distinctive phases in the propagation of each fire…" – what was the basis for assuming a 50:50 split of AGS between phase 1 and 2, and 25:75 split of BGS between phase 2 and 3? I notice also that Figure 4 doesn't seem to show much trend of decreasing MCE with time for any of the fires – even BIS – which would seem to call into question this assumption about the evolution of the combustion phases. This should be commented upon – it would be nice to see either on the same plots in Figure 4, or else as a separate figure to compare with, what the time evolution of MCE looks like for the authors' bottom-up estimation of CO2 and CO emissions. Currently only the time-mean value of MCE is compared with the tower observations, but comparison of the daily time evolution of predicted MCE with the atmospheric tower observations will help to validate the assumption that the fire transitions from fully flaming to fully smouldering.

 **table 4 provides MCE for the 3 stages of combustion namely spreading, mixed and post spreading. we do simulate a decrease in MCE across stages, as observed during the 4 first days of the BIS fire down to a low value of 0.79 observed and 0.730 simulated. MCE increases the 5th day at BIS from 0.82 to 0.85. Soil temperature maximum in soils might increase CO emissions**

**during day 199 when more than 80% of the final burned area was already covered, date after which the fire started to decrease in intensity (figure 5).**

**our representation of spreading, mixed and post-spreading phase represent more full flaming, mixed flaming/smoldering and full smoldering combustions to capture the min and max MCE observed at the tower, rather than a direct time series of emissions. Many other factors as fire intensity and fire rate of spread might influence these values at the tower level, that we don't capture in our emissions values. it would be very adventurous to try and capture the actual temporal trend of MCE.**

**regarding considering 50:50 for flaming between the spreading and mixed stages, and 25:75 for the smoldering between mixed and post spreading stages, it results from a combination of both instantaneous emissions ratios between flaming and smoldering and fraction of AGS and BGS consumed during these stages. the fire spread stage is only flaming (100%) thus affecting AGS. the post fire stage is only smoldering (100%) thus affecting only BGS. as we want to calculate CO and CO2 emissions over the whole burned area we had to solve the following equation:**

**$a*F + (1-a)F + (1-b) * S + b*S$ = Emissions with MCE mixed= $((1-a)$ MCEspreading + $(1-b)$ MCEsmoldering) / $(1-a+1-b)$**

**As we investigated from hotspots that smoldering lasted longer (15 days after the end of spreading to be conservative) than the spreading phase (10 days), we considered that the mixed phase (half of the 10 day spreading =5 days), was 25% of the smoldering period covering 15days only smoldering + 5 days mixed (=20days). when considering this value we reached MCE in accordance with measurements at the tower. using 20:80 or 10:90 would have increased the MCE during the mixed stage out of the measurement at the tower, but would not have changed the total emissions. We understand this choice was not fully justified, and could be subjective, but we managed to adjust our emissions to MCEs and smoldering duration from observations. we expanded the explanation of this choice in the section 2.5.**

L338: "biome-specific standard emission factors (in kgDM.MJ-1)" – what makes them 'standard'? In fact, GFAS I think is somewhat unique in taking this FRP-based approach – other inventories such as GFED take an approach similar to the authors', of relating fuel consumption to the burned area and a map of vegetation. Also, the biome-specific kg[DM]/MJ in GFAS are not emission factors, they are the dry matter combustion rate. These are used to estimate the amount of DM combusted. The emission factors are then the mass of CO2 or CO emitted per DM combusted.

**thanks for this clarification. we updated the text accordingly.**

L369: "Daily MCE variations (Fig. 4) emphasized a decreasing trend for the BIS fire" – this decreasing trend is quite marginal, and I wouldn't describe it as being emphatic in any sense. Over the 5 days shown on Figure 4, the BIS daily median MCE goes up, down, up, up – and ends up back close to its initial value. So it's a weak decreasing tendency at best. I'm also confused because elsewhere the authors state that both the BIS and ROC fires lasted for more than 25 days, and yet in Figure 4 the MCE is plotted only for 5 and 3 days respectively. It would be nice to see whether over the full 25 day+ duration of the event, there is more of a trend towards lower MCE after the first few days.

**The calculation of the MCE from ground towers did not produce a complete time-series for us to be able to track the variations of the fires over a longer time period. The data presented in Figure 4 corresponds to the data where robust estimates are obtained considering the Lagrangian influence selection criteria and the background estimates generated by the Random Forest model when the plume was actually reaching the tower. In turn, the fire flaming duration is represented by hotspots (figures 5 and A3) and lasted 10 days for BIS and 2 days for ROC,**

**and fire duration including post spreading smoldering captured by remaining hotspots lasted up to 17days for ROC and more than 25days for BIS.**

**L422-423:** "we conclude that fires prone to experiencing smoldering combustion, such as BIS and ROC fires, exhibit a prolonged duration of hotspots after ignition" – I would agree that prolonged duration of hotspots after ignition could be an indication that smouldering combustion is happening, however it's not a pre-existing property of the vegetation or something that can be observed in the initial fire behaviour, so I'm not clear how it could be used to indicate beforehand that a fire is prone to smouldering combustion.

 **definitely true, this index cannot be used for a near-real time fire emission assessment (GFAS approach) but an a posteriori one (GFED approach). we inserted that comment in the text section 3.3.**

**L436-437:** "aligning with the median value obtained from the hourly mixing ratios measured at the ROC tower" – this was previously reported as a mean value (L366) – which is correct? This sentence is also inconsistent with L432-433 where it was suggested that ABS-only MCE diverges from that measured at ROC. If flaming-only MCE aligns closely with the average reported for ROS, does that mean that overall, smouldering contributed little to the total emissions for this fire, and it was only BIS where smouldering-only emission factors were really necessary to get the correct carbon budget?

In fact, it's the median values that are referenced here. Corrections have been made accordingly.

We have modified the sentence as follows.

While these values closely mirrored the MCEs observed at the OHP tower, they notably diverged from the range of MCEs captured at the ROC and BIS stations.

**L449:** "Drawing from our MCE-derived carbon emissions estimates of AGS-BGS combustion" – how was MCE used to derive the carbon emission estimates? My understanding was that these were derived independently, using the carbon pools estimated by land cover maps, burnt pixels from hotspot detection, and literature values of combustion completeness, emission factors etc. It would be really cool if the observed MCE could be used to derive the carbon emissions directly, but that doesn't seem to have been what was done here. Unless I've misunderstood!

 **we modified the text as we developed a carbon emission framework based on the witnessing of, and the quantification of, soil smoldering in MCE measured at the tower, not a direct MCE-derived calculation as tower-MCEs are discontinuous in time (cf previous comment for L369) and not available nor yet easily automatized to filter out days not influenced by fire plumes.**

**L469-470:** "the GFAS framework estimated that summer fires were accountable for 3.86 MtCO2 emissions when excluding belowground combustion" – again, I would say that technically GFAS doesn't exclude belowground combustion, since it has biome-specific fuel consumption and/or emission factors for 'peat' and 'extratropical forest with organic soil' which in principle should reflect the different carbon stock and the balance of smouldering versus flaming fires. However, it's quite likely the course resolution land cover maps GFAS uses don't categorise these areas of France as belonging to either of those biomes. So, I would view one of the major take-home messages as being that your emissions estimate is only as good as your fuel cover data.

**We agree that GFAS can implement soil smouldering and we rectified properly this point all along the manuscript. "Peat" and "Extratropical forest with organic soil" (EFOS) and other classes with organic soils (XXOS) are however not referenced in temperate forest. The goal of the study is to address soil combustion in such forest. our major take home message is that extratropical forests with organic soil might not be well defined yet, or should be extended to temperate and mid latitude forests and that we demonstrate with MCE that they actually burn**

**with smouldering combustion. National emission inventories would then benefit for further assessment of potential soil fires mapping, and referencing fine resolution peatlands. this could lead to substantial revision of the fire carbon emissions. we agree that the combustion process we used is similar to other assessment tools and we benefited from higher resolution information, and in addition a data driven demonstration from MCE that these fires happen.**

**Sect. 3.5, Figure 6, and Table 4:** These all show uncertainty ranges for the various DM and emission values, however as far as I can see no description is given anywhere (in the Methods, in the Results, or in the Figure captions) as to what this uncertainty represents (e.g. is it ±1σ, 95% confidence interval, something else…) or how it was calculated. (Apologies if it is described somewhere and I've missed it).

**It was indeed not mention before. See response to comment on L321**

**L542-544:** "To maintain a conservative approach, we adopted a ROS of 0.2 cm.h-1 for soil combustion, resulting in a daily consumption of approximately 4.8 cm" – shouldn't this have been mentioned in the Methods, if this calculation determines how deep the soil layer is burned? (C.f. my previous comment; it's not made clear what value is used for the combustion completeness, which can dramatically alter the resulting carbon emissions calculation).

**cf next comment**

**L543-545:** "we adopted a ROS of 0.2 cm.h-1 for soil combustion, resulting in a daily consumption of approximately 4.8 cm, which roughly corresponds to 40 cm burned over an 8-day period, which corresponds to the average flaming duration of our fires. This 40 cm of consumed peat aligns with the upper bound of our soil combustion parameters" – this seems a little concerning. The BIS and ROC fires last much longer than 8 days, and yet after 8 days the model assumes that you have already burned close to the upper bound of the maximum peat depth that can be burned. Most real-world peat fires do not burn nearly as deep as 40cm. This might imply that the authors' model is typically assuming too high belowground DM combusted, by predicting unrealistically high depths of burn.

**the CC of peatland fires is referenced in table 4, with min value being 0.05 and maximum value 0.2. As described in the peatland carbon description section 2.4.5 we assumed a peatland depth of 2m, so that burn depth varies between 0.05*200cm=10cm and 0.2*200cm=40cm. our mean value is then 25cm. we explicitly mentionned in the text that 40cm was the maximum value, but we rephrased section 4.2 to better clarify that point.**

**L549-551:** "For a comparative perspective, 550 Mickler et al. (2017) using fine resolution LIDAR data revealed that peatland wildfires could exhibit an average burn depth of 42 cm" – this was a case study of a single fire – it cannot be interpreted to mean that 42cm is typical of most peat fires. As the authors have already noted, peat fires can burn anything from 1 cm to 50cm+, but in field studies most are < 20cm (see e.g. Walker at al (dataset, 2020) https://doi.org/10.3334/ORNLDAAC/1744 or Blackford et al. (2024) https://doi.org/10.5194/gmd-17-3063-2024)

**cf comment above.**

**L551:** "average belowground carbon emissions estimated at 544.43 t C ha−1" – the depth of burn in Mickler et al. of 42cm is almost exactly the same as the depth the authors report here of 40 cm. So why is the carbon emitted per ha so much higher in the Mickler et al. study?

our estimate of 172tC.ha-1 is actually for the mean burn depth of 25cm, with +/- 74 tC.ha-1 when considering the uncertainty varying between 10 and 40cm. the only difference between the two studies come from the soil organic matter content and the bulk density. We used local data of 145 kgDM.m-3, as measured in France (Pilloix, 2019). I could not find this information in Mickler et al.

**L571:** "This enables our bottom-up approach to be confronted and evaluated against atmospheric MCEs" – indeed – this is why it would be really good if the MCE predicted from the authors' bottom-up approach could also be plotted side-by-side with the MCE observed at the tower stations, e.g. in Figure 4, or with an extra figure straight after Fig. 4 that has the same plots but for the bottom-up estimate MCE.

**We add a new plot showing MCE obtained from the fire emission model and referenced in Table 4. This new plot allow a better comparison of tower-measured MCE and MCE obtained from the model**

**L604:** "These low MCE values, which are challenging to account for based on biomass or SOM combustion alone" – how low, and can the authors demonstrate that these values require a contribution from lignite combustion to account for? (N.B. not sure MCE values for lignite are ever described in the text, making it hard to judge).

**=>we updated table2 with MCE calculations for each carbon pool, so that MCE lignite = 0.66 while MCE SOM=0.79. From this information, we illustrate that the lowest MCE which can be reached without accounting for lignite combustion is 0.79 while measurements reached lower values. In turn, only accounting for lignite combustion could lower this value.**

**L692-695:** "Finally, we advocate for the widespread use of our updated fire emissions processing chain for France, which could potentially be extended to other European temperate forests" – to try and motivate this further, along with the argument in Sect. 4.4 that these fires were a larger contribution to France's carbon emissions than previously anticipated: is there any validation that can be done to confirm the accuracy of your CO2 and CO total emission amounts? This feels like one weakness of the bottom-up approach here; it's based on a lot of different input parameters from the literature (e.g. land cover type, carbon density, combustion completeness etc.) all of which have large uncertainties in them (as mentioned previously, the uncertainty range in the BGS combustion completeness alone could potentially explain the difference with the GFAS emission budget). The manuscript has shown that the relative CO2/CO ratio is *consistent* with the MCE observed at the tower stations, but are there any measurements (from the tower observations or otherwise) that can be used to additionally constrain the total emission amount, to validate that your approach gives the correct total magnitude whereas GFAS is definitely too low? Otherwise, either could be right for the total magnitude (even if GFAS probably gets the wrong MCE).

**there is no such large scale measurements of CO or CO2 to validate any of the current emission estimates, locally or globally. I understand the reviewer concern that our estimates could be as wrong as the other. However, by using similar CC and EF as other emission processings (GFED or GFAS), we only tested the MCE information which could not be reached with current pools combusted for temperate forests. Our approach solved this issue, and concluded for an increased total emission. Our conclusion is then to push toward better identifying soil combustion in temperate ecosystems, a better ampping of peatlands and raise the attention toward unsuspected potential carbon pools able to be exposed to fire as superficial charcoal layers or coal mine residues piled up in the 19/20th century in some northern Europe area. we rephrased the conclusion to better capture this message.**

**Technical comments:**

**L15-16:** "coarse remotely-sensed burned area data… to provide near real-time information" – burnt area-based products, such as GFED or FireCCI, are not normally near real-time. Near real-time products, such as GFAS, use fire radiative power or active fire counts to achieve this.

**we removed 'to provide near real time information'.**

**L32-33:** "the generic GFAS global estimates" – what do the authors mean by this? The authors are comparing emissions from France, so presumably the value from GFAS can't be global. What makes it 'generic'?

**we rephrased this sentence.**

**L49:** "limited information is currently accessible" – do the authors mean accessible, or available? I.e. does the information exist but is not easy to get access to, or does the data simply not exist?

**I would say both for fire data, as some ancient data on fire registration in northern france might exist, but are hardly … accessible or have been lost or stored we don t know where. Anyway, i believe 'available' fits better. we modified accordingly.**

**L71:** "firefighters consistently raised concerns about lingering soil fires…" – please add citation if possible

**I was referring to newspaper/tv information in French. here is the link to newspaper note on august 12th 2022 for example warning about soil zombie fires lighting new fire ignition in Landiras (Landes pine forest). we inserted this note in the text.**

**https://www.ouest-france.fr/faits-divers/incendie/feux-zombies-a-l-origine-de-la-reprise-des-incendies-en-gironde-on-vous-explique-ce-phenomene-00749e06-1a38-11ed-9b31-1adf573d9c14**

**L73:** "wash away the burning soil material" – perhaps these fires were unusual, but are smouldering fires typically extinguished by the burning material being washed away through runoff? Normally I'd assume it would be extinguished just through fully saturating the soil – at least for peaty soils, and especially where the fire has burnt down into the soil and/or is smouldering underground (one reason why it requires such large volumes of water to extinguish them, as the peat can hold a lot of moisture before combustion is inhibited, and can continue to smoulder below the surface layer meaning that the entire soil column needs to be saturated). Rainfall may erode the already burnt fine ash, of course, but a priori it seems like you'd need really a lot of soil erosion for this mechanism to extinguish an active smouldering fire that's occurring any significant depth into the soil.

**we removed this part of the sentence. i guess we mis-used the term 'wash way'.**

**L87-89 and L90-92:** "Various studies of smoke chemical analysis… have determined MCE indices ranging from 0.6 to 0.8 during smoldering combustion" and "Hu and Rein (2022) recently compiled a review on smoldering combustion emission factors, with MCE indices varying from 0.93 for flaming in forests to 0.85 for peatland smoldering combustion" – these statements seem inconsistent. How can the literature-average value of 0.85 in Hu and Rein, be outside of the range reported by 'various studies'?

**the Hu and Rein analysis actually estimate a mean value of 0.85, ranging between 0.7 and 0.9. our 'various' are just a few ones we picked up in the literature, not being exhaustive. we now inserted the range of values**

**L163-164:** "a spatial filtration process to exclude all thermal anomalies… corresponding to non-forest fires" – how did the authors spatially filter to exclude non-forest fires? Presumably using a high-resolution map of land cover type, in which case the source should be cited here. Was the map updated and current for 2022? Or, if complied before 2022, could forested areas have changed over time?

**we actually made it more simple… we cropped the fire hotspot data within the fire polygons. every hotspot outside the fire polygones were not considered, whatever they are. for clarity, we remove this last part of the sentence.**

**L100:** "variations in atmospheric MCE" – technically the MCE is a property of the combustion, not of the atmosphere. Its value can be estimated from atmospheric measurements, but the atmosphere itself doesn't have a combustion efficiency (well, not at normal temperatures/pressures anyway)!

**We replaced "atmospheric MCE" by tower-measured MCE"**

**L159-160:** "we harnessed VIIRS data from the SNPP and NOAA sources" – The authors name the other satellite missions (Aqua, Terra, S-NPP) and so for consistency should name the second VIIRS satellite as well rather than just saying NOAA, who operate many satellites. The satellite with VIIRS onboard is NOAA-20.

**Right, replaced**

**L172:** "Fig. 5" – this figure should have been numbered as Fig. 2

**Figures numbers have been corrected according to previous comment**

**L218-219:** "The meteorological data encompassed parameters such as…" – 'such as' is insufficient; please give a complete list of the variables that were used as predictors in the random forest model.

**The complete list of parameters used in the random forest model is the one mentioned in the text. Thus, the word 'such as' was removed from the sentence.**

**L234:** "we sought to estimate the pools" – carbon pools?

**Replaced**

**L291-292:** "The bulk density of brown coal generally hovers around 700kgDM.m-3" – 'generally hovers around' to me implies something that fluctuates over time (e.g. stock prices, exchange rates, etc.), which I'm not sure is the case for bulk density of coal (except on geological timescales, I suppose).

**True, we replaced "generally hovers around" by "is generally around"**

**L297-298:** "we quantified CO2 and CO emissions" – 'estimated' might be a better term, to make it clear that (as I understand it) the CO2 and CO emissions are not directly measured or constrained, but rather inferred bottom-up based on estimated carbon pools and some assumptions of combustion completeness, smouldering fraction etc.

**You are right, we replaced "quantified by estimated"**

**L318:** "the entire fire (F)" – 'F' has already been used previously to mean "flaming (F)" (L298), making the $E_{Fx}$ notation ambiguous.

**True, we replaced (fire (F) by fire (A) in both text and equation**

**L363-364:** "The BIS site shows mostly low minimum values" – what counts as 'low'?

**The word low refers in this case to the MCE value used to distinguish between blazing and smoldering fires. To avoid the confusion the sentence was changed as follows:**

> *The BIS values correspond to the MCE values that are observed most often under smoldering combustion phases and high-temperature pyrolysis phases.*

**L367:** "ROC exhibited minimum values that reached 0.82" – Figure 4 seems to show hourly MCE values that are < 0.8, which doesn't seem to match up with the text here.

**In figure 4, the top left boxplots corresponds to the 1-hr MCE. And the minimum value of the outlier corresponds to 0.83. However, in the case of the of the 1-min MCE, the calculated MCE went below 0.8.**

**L367:** "far beyond" -> 'far lower than'?

**Replaced.**

> *However, ROC exhibited minimum values that reached 0.82, far lower than the values observed at OHP.*

**L431-433:** "The resulting MCEs ranged from 0.955 to 0.961 for all the fires, with no significant distinctions between them. While these values closely mirrored the MCEs observed at the OHP tower, they notably diverged from the MCEs captured at the ROC and BIS stations" – the mean MCE for ROC was 0.94, so it's not really diverging very far from this station either, surely?

**=>ROC might have a median MCE of 0.94, but with a wide range varying from 0.75 to 0.95, while OHP experiences a way lower variation. we rephrased and emphasized these extreme values observed in ROC and BIS.**

**L455-456:** "Fires mainly altered forest areas in the Atlantic pine region (76.5%) and other forest (75.6%) regions" – I'm not clear what these percentages represent. It seems to say that 77% of fires have been in Atlantic pine regions, while 76% of fires have been in other forest regions? Which doesn't make sense – maybe a typo?

**This was indeed a typo (the sum of both Atlantic pine forest and other forest area). This sentence has been removed as it disrupt the logical continuation of the paragraph**

**L488:** "soil vegetation" -> 'soil and vegetation'?

**Replaced**

**L548-549:** "172 (± 74) tC.ha-1 emitted, which is slightly higher than the value of 96tC/ha" – 'slightly higher' is kind of an understatement here. 172 is a lot higher than 96.

 true indeed. we removed 'slightly'.

**L570:** "the existing generic fire emissions assessments" – I'm still not clear what makes them 'generic'

**we removed 'generic'. our goal was to say here that GFAS was a standard and widely used dataset globally and considered as a reference to compare with.**

**L591:** "excludes these processes" – again, not strictly true, since GFAS has a 'peat' land classification with a different fuel consumption rate and CO2/CO emission factors, and a 'Extratropical forest with organic soil' land classification with a higher fuel consumption rate than regular extratropical forest. (Though I certainly agree that it's only included in a very crude way, and the authors' method is far more sophisticated! And of course lignite is not included, and the contribution from lignite here is a very important conclusion to highlight).

**"Peat" and "Extratropical forest with organic soil" (EFOS) and other classes with organic soils (XXOS) are not present in temperate forest. The goal of the study is to address soil combustion in such forest. Thus, we added "in temperate forest**

**L647:** "might be actually true" -> 'might also be true'?

**"might be actually true" has been changed to "might be true"**

**L662:** "lower" -> 'low'

**"lower" has been changed to "low" on line 662**

**Figure 2:** Axis labels on the contour bars are much too tiny to be readable!

**As the figures are intended to show an example of the fire emission model's application, the values are of no real interest. Consequently, the captions have been removed and the figure updated.**